# Dynamics of Spontaneous Topic Changes in Next Token Prediction with Self-Attention

**Mumin Jia**[*]
Department of Mathematics and Statistics
York University
Toronto, Ontario M3J 1P3
amyjia@yorku.ca

**Jairo Diaz-Rodriguez**[*]
Department of Mathematics and Statistics
York University
Toronto, Ontario M3J 1P3
jdiazrod@yorku.ca

## Abstract

Human cognition is punctuated by abrupt, spontaneous shifts between topics—driven by emotional, contextual, or associative cues—a phenomenon known as spontaneous thought in neuroscience. In contrast, self-attention-based models rely on structured patterns over their inputs to predict each next token, lacking spontaneity. Motivated by this distinction, we characterize *spontaneous topic changes* in self-attention architectures and reveal divergences from *spontaneous human thought*. First, we establish theoretical results under a simplified, single-layer self-attention model with suitable conditions by defining a topic as a set of Token Priority Graphs (TPGs). Specifically, we demonstrate that (1) the model maintains the priority order of tokens related to the input topic, (2) a spontaneous topic change can occur only if lower-priority tokens outnumber all higher-priority tokens of the input topic, and (3) unlike human cognition, the longer context length or the more ambiguous input topic does not increase the likelihood of spontaneous change. Second, we empirically validate that the effect of input length or topic ambiguity persists in modern, state-of-the-art LLMs, underscoring a fundamental disparity between human cognition and AI behavior in the context of spontaneous topic changes. To the best of our knowledge, no prior work has explored these questions with a focus so closely aligned to human thought.

## 1 Introduction

Human cognition is punctuated by abrupt, apparently unstructured topic changes, the hallmark of *spontaneous human thought*, a phenomenon that has become a central topic in cognitive neuroscience [4, 8, 9, 23, 32–34]. For example, a spontaneous shift in focus during a conversation, a sudden leap between ideas when brainstorming, or an unexpected redirection in storytelling. These abrupt changes may be due to an emotional connection, such as recalling reading a book during a family vacation, where sensory details like the scent of the ocean or the warmth of the sun trigger a vivid memory. However, LLMs shift topics in response to contextual cues in the input, rather than initiating *spontaneous topic changes* on their own. They follow a structured, statistical approach, remaining on topic unless explicit cues signal a change. Figure 1 illustrates this distinction using the first sentence of the book "One hundred years of solitude" [11].

Our work takes initial steps toward formalizing the dynamics of *spontaneous topic changes* in LLMs and analyzing how they relate to or diverge from *spontaneous human thought*. To this end, we ground our theoretical analysis in a single-layer self-attention model and empirically extend it to modern LLMs, laying the groundwork for drawing comparisons between AI models and human cognition.

---

[*]Equal contribution.

39th Conference on Neural Information Processing Systems (NeurIPS 2025).

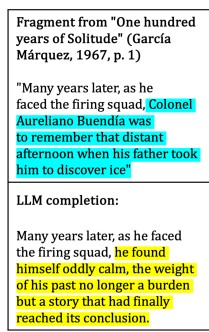

Figure 1: Illustration of the difference between human cognition and LLMs. The original fragment of "One hundred years of solitude" [11] (**top**) has a clear spontaneous thought, but the GPT-2's completion (**bottom**), demonstrates continuity.[2]

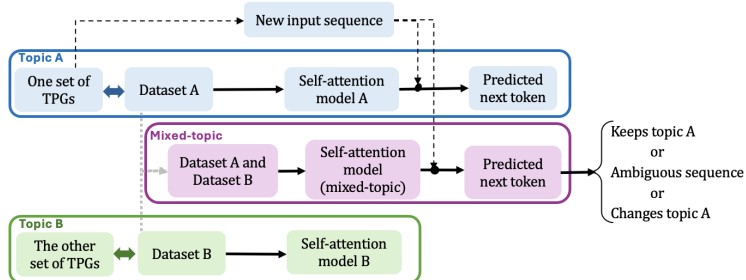

Figure 2: **Overview of our theoretical framework.** We define a topic as a set of TPGs $\{\mathcal{G}^{(k)}\}_{k=1}^{K}$ (Def. 2) and generate a dataset for each topic. The combination of dataset A and dataset B becomes the dataset for the mixed-topic model. We train self-attention models independently on each dataset. Then, we generate a new input sequence from topic A and predict the next token with two models, self-attention model A and self-attention model (mixed-topic). The next-token prediction with the mixed-topic model is categorized into three outcomes: keeps topic A (*topic continuity* from Def. 3); *ambiguous sequence* (from Def. 4); or changes topic A (*change of topic* from Def. 5). Further details for each category are shown in Figure 3.

Recent advancements in the related field have substantially deepened our understanding of self-attention architectures. Li et al. [25], Tarzanagh et al. [48, 49] have linked the self-attention to support vector machines (SVMs), offering optimization strategies for next-token prediction. Li et al. [26] highlight that in mixed-topic inputs, transformers achieve higher pairwise attention between same-topic words compared to different-topic words. In parallel, prior studies have recognized the practical challenges of *spontaneous topic changes* in LLMs and proposed approaches to address them [19, 27, 28, 36, 46, 55]. Notably, *spontaneous topic changes* must be differentiated from hallucinations, generating incorrect or fabricated information without a clear contextual basis [20, 31].

Despite these advancements, our understanding of the dynamics of *spontaneous topic changes* in LLMs remains limited. Investigating the relationship between *spontaneous topic changes* in self-attention models and *spontaneous human thought* can provide valuable insights into the cognitive discrepancies of current language models compared with humans. Since modern LLMs rely on self-attention architectures, we begin by theoretically characterizing *spontaneous topic changes* in a simplified setting. We then extend these findings through experiments on more complex, state-of-the-art models. To the best of our knowledge, no prior studies have investigated these dynamics so closely in relation to human thought.

Figure 2 outlines our theoretical framework. To make the mathematical analysis tractable, we follow the same single-layer self-attention framework with log-loss objective function governed by Assumptions 1–4 from Li et al. [25]. Inspired by token-priority graphs (TPGs) [25] and building on attribution graphs from Ameisen et al. [1] for exposing an LLM's internal computation, we define a topic as a set of TPGs. This graph-based formulation aligns naturally with recent advances in structured representations for LLMs [42, 52]. Furthermore, this mirrors neuroscience models of spontaneous human thought, in which concepts serve as nodes connected by associative edges [32]. Despite relying on these specific settings, our experiments extend our findings to modern LLMs, empirically confirming that relaxing these assumptions does not seem to undermine our core insights.

## 1.1 Summary of findings

Imagine an oracle that is an expert on Topic A, capable of following any conversation within that topic while staying true to its context. Now, suppose the oracle gains knowledge of Topic B and is following a conversation about Topic A. Will the oracle's responses remain within Topic A, or will

---

[2]Just to illustrate, we use the prompt *Please continue this short sentence, forgetting about "One hundred Years of Solitude"*, since on a real conversation the LLM would be blind to the final output.

the influence of the knowledge of Topic B cause the conversation to drift? This analogy encapsulates the problem we address: understanding when and why attention models might preserve a topic or change to another spontaneously. Specifically, we make the following contributions:

1. **Preservation of input topic priorities**. Using a controlled sandbox, we demonstrate in Theorem 2 that self-attention models trained on mixed-topic datasets maintain the priorities of tokens associated with the original topic of an input sequence (Topic A in our analogy).

2. **Changing topics triggered by token frequency**. In Theorem 3, we show that the oracle's responses may reflect a change of topic only if a lower-priority token appears more frequently than all higher-priority tokens of Topic A.

3. **Impact of input length and topic ambiguity**. Theorem 4 establishes that longer input sequences decrease the likelihood of changing topics. Furthermore, input topic ambiguity acts as a stabilizing factor, not increasing the frequency of spontaneous topic changes.

4. **Difference between LLMs and human cognition**. In Section 6 we empirically extend Theorem 4 to modern, deeper LLMs. Unlike human cognition, where extended discussions often encourage spontaneous thoughts and topic ambiguity promotes cognitive connections, our results highlight the opposite behavior in LLMs: neither longer prompts nor greater topic ambiguity appreciably increases the likelihood of a spontaneous topic change.

**Overview of the paper structure.** We begin with the problem setup in Sec 2. Sec 3 introduces the definition of topic, and Sec 4 examines how self-attention models allocate the token priorities within the mixed topics. In Sec 5, we establish the conditions under which a self-attention model induces *spontaneous topic changes* and show the dynamics of topic changes with longer input sequences or the presence of topic ambiguity. We then extend our analysis to frontier LLMs in Sec 6. Related work and discussion are provided in Secs 7 and 8, respectively. All proofs are provided in Appendix A.

## 2 Problem setup

### 2.1 Next topic prediction with self-attention model

In line with the approach presented by Tarzanagh et al. [49] and Li et al. [25], we frame the next-token prediction task as a multi-class classification problem. Given a vocabulary of size $K$ with an embedding matrix $\mathbf{E} = [\mathbf{e}_1 \ \mathbf{e}_2 \ \cdots \ \mathbf{e}_K]^\top \in \mathbb{R}^{K \times d}$, we aim to predict the next token ID $y \in [K]$ based on an input sequence $\mathbf{X} = [\mathbf{x}_1 \ \mathbf{x}_2 \ \cdots \ \mathbf{x}_T]^\top \in \mathbb{R}^{T \times d}$ with $\mathbf{x}_i \in \mathbf{E}$ for all $i \in [T]$. The training dataset, denoted as

$$\text{DSET} = \{(\mathbf{X}_i, y_i) \in \mathbb{R}^{T_i \times d} \times [K]\}_{i=1}^n,$$

contains sequences of varying lengths $T_i$. In our notation $\mathbf{x}$ is the embedding vector corresponding to the token ID $x$, this is $\mathbf{x} = \mathbf{e}_x$. For prediction, we utilize a single-layer self-attention model with a combined key-query weight matrix $\mathbf{W} \in \mathbb{R}^{d \times d}$ and identity value matrix as in Tarzanagh et al. [49]. The self-attention embedding output

$$f_{\mathbf{W}}(\mathbf{X}) = \mathbf{X}^\top \mathbb{S}(\mathbf{X}\mathbf{W}\bar{\mathbf{x}}), \tag{output}$$

where $\mathbb{S}(\cdot)$ is the softmax operation and $\bar{\mathbf{x}} := \mathbf{x}_T$, serves as a weighted representation of the tokens, allowing for context-sensitive prediction of $y$ based on the final input token. Let $\ell : \mathbb{R} \to \mathbb{R}$ be a loss function. For the training dataset DSET, we consider the empirical risk minimization (ERM) with:

$$L(\mathbf{W}) = \frac{1}{n} \sum_{i=1}^n \ell(\mathbf{c}_{y_i}^\top \mathbf{X}_i^\top \mathbb{S}(\mathbf{X}_i \mathbf{W} \bar{\mathbf{x}}_i)). \tag{ERM}$$

We assume a well pre-trained classification head matrix $\mathbf{C} = [\mathbf{c}_1 \ \mathbf{c}_2 \ \cdots \ \mathbf{c}_K]^\top \in \mathbb{R}^{K \times d}$. Each classification head $\mathbf{c}_k \in \mathbb{R}^d$ is fixed and bounded for all $k \in [K]$. Starting from $\mathbf{W}^{(0)} \in \mathbb{R}^{d \times d}$ with step size $\eta > 0$, for $\tau \geq 0$ we optimize $\mathbf{W}$ with a gradient descent algorithm

$$\mathbf{W}^{(\tau+1)} = \mathbf{W}^{(\tau)} - \eta \nabla L(\mathbf{W}^{(\tau)}). \tag{Algo-GD}$$

We keep the first two assumptions from Li et al. [25]:

**Assumption 1.** $\forall y, k \in [K], k \neq y, \mathbf{c}_y^\top \mathbf{e}_y = 1$ *and* $\mathbf{c}_y^\top \mathbf{e}_k = 0.$

**Assumption 2.** *For any* $(\mathbf{X}, y) \in$ *DSET, the token* $\mathbf{e}_y$ *is contained in the input sequence* $\mathbf{X}$.

Assumption 1 represents a variation of the weight-tying approach commonly used in language models [40, 50]. Once training is complete, for a new input sequence $\mathbf{X}$, and a model characterized by $\mathbf{W}$, we predict the next token ID $\hat{y}_{\mathbf{w}}$ based on greedy decoding the probabilities from the softmax of the classification output

$$\hat{y}_{\mathbf{w}} \in \arg \max_{k \in [K]} \left[ \mathbb{S} \left( \mathbf{C} f_{\mathbf{W}}(\mathbf{X}) \right) \right]_k. \tag{1}$$

## 2.2 Token-priority graph and global convergence of the self-attention model

Li et al. [25] defined a *token-priority graph (TPG)* as a directed graph with nodes representing tokens in the vocabulary. $\text{DSET}^{(k)}$ is a subset of sequences from DSET with the same last token is $\mathbf{e}_k = \bar{\mathbf{x}}$. They defined TPGs $\{\mathcal{G}^{(k)}\}_{k=1}^K$ such that every $\mathcal{G}^{(k)}$ is a directed graph where for every sequence $(\mathbf{X}, y) \in \text{DSET}^{(k)}$ a directed edge is added from $\mathbf{e}_y$ to every token $\mathbf{x} \in \mathbf{X}$. TPGs are further divided into *strongly-connected components (SCCs)*, which capture subsets of tokens with equal priority. For tokens within two different SCCs, strict priority orders emerge, helping the model to differentiate between tokens when learning next-token predictions. We use the same notation as Li et al. [25], given a directed graph $\mathcal{G}$, for $i, j \in [K]$ such that $i \neq j$:

- $i \in \mathcal{G}$ denotes that the node $i$ belongs to $\mathcal{G}$.
- $(i \Rightarrow j) \in \mathcal{G}$ denotes that the directed path $(i \to j)$ is presented in $\mathcal{G}$ but $j \to i$ is not.
- $(i \asymp j) \in \mathcal{G}$ means that both nodes $i$ and $j$ are in the same strongly connected component (SCC) of $\mathcal{G}$ (there exists both a path $i \to j$ and $j \to i$).

For any two distinct nodes $i, j$ in the same TPG, they either satisfy $(i \Rightarrow j)$, $(j \Rightarrow i)$ or $(i \asymp j)$. Nodes in each $\mathcal{G}^{(k)}$ represent indices in $[K]$, and SCC structure supports the self-attention mechanism's ability to assign priority within sequences based on the conditioning last token. Theorem 2 of Li et al. [25] proved that under Assumptions 1 and 2, the self-attention model learned through Algo-GD converges to the solution of the following Support Vector Machine (SVM) defined by the TPGs of the underlying dataset DSET

$$\mathbf{W}^{\text{svm}} = \arg \min_{\mathbf{W}} \|\mathbf{W}\|_F \tag{Graph-SVM}$$

$$\text{s.t.} \quad (\mathbf{e}_i - \mathbf{e}_j)^\top \mathbf{W} \mathbf{e}_k \begin{cases} = 0, & \forall (i \asymp j) \in \mathcal{G}^{(k)} \\ \geq 1, & \forall (i \Rightarrow j) \in \mathcal{G}^{(k)} \end{cases} \forall k \in [K].$$

Here is a condensed version of the theorem:

**Theorem 1** (Li et al. [25]). *Consider dataset DSET and suppose Assumptions 1 and 2 hold. Set loss function as* $\ell(u) = -\log(u)$. *Starting Algo-GD from any* $\mathbf{W}(0)$ *with constant size* $\eta$, *if* $\mathbf{W}^{\text{svm}} \neq \mathbf{0}$,

$$\tilde{\mathbf{W}} = \lim_{\tau \to \infty} \frac{\mathbf{W}(\tau)}{\|\mathbf{W}(\tau)\|_F} = \frac{\mathbf{W}^{\text{svm}}}{\|\mathbf{W}^{\text{svm}}\|_F} \tag{2}$$

This convergence implies that the model predicts the next token based on priorities obtained from the SCCs within the TPG relevant to the last token of the input sequence. Unlike the work in Li et al. [25], which considers both hard retrieval and soft composition components and examines multiple loss functions in subsequent results, we focus exclusively on a log-loss function in this work, leaving the exploration of other loss functions for future research. Since the soft composition component is not required for our subsequent definitions and theoretical results, we concentrate solely on the hard retrieval component.

We add here another reasonable assumption that prevents the probabilities in Equation 1 from being equal due to improbable numerical reasons, and we present our first lemma.

**Assumption 3.** *For any* $(\mathbf{X}, y) \in$ *DSET,* $\exists i, j \in [T]$ *and* $u, v \in \mathbb{Z}$ *such that* $u \left[ \mathbb{S}(\mathbf{X} \tilde{\mathbf{W}} \bar{\mathbf{x}}) \right]_i = v \left[ \mathbb{S}(\mathbf{X} \tilde{\mathbf{W}} \bar{\mathbf{x}}) \right]_j$ *if and only if* $u = v$ *and* $\left[ \mathbb{S}(\mathbf{X} \tilde{\mathbf{W}} \bar{\mathbf{x}}) \right]_i = \left[ \mathbb{S}(\mathbf{X} \tilde{\mathbf{W}} \bar{\mathbf{x}}) \right]_j$.

**Lemma 1.** *Suppose conditions from Theorem 1 and Assumption 3 hold. Consider an input sequence* $\mathbf{X}$ *from* $\text{DSET}^{(k)}$ *and corresponding TPG* $\mathcal{G}^{(k)}$, $\forall i, j \in [K]$ *we have* $\left[ \mathbb{S} \left( \mathbf{C} f_{\tilde{\mathbf{W}}}(\mathbf{X}) \right) \right]_i = \left[ \mathbb{S} \left( \mathbf{C} f_{\tilde{\mathbf{W}}}(\mathbf{X}) \right) \right]_j$ *iff* $(x_i \asymp x_j) \in \mathcal{G}^{(k)}$.

This means that the tokens that maximize the probability for weights $\tilde{\mathbf{W}}$ in Equation 1 are all within the same SCC leading to the following definition:

**Definition 1** (*highest probability SCC*). *Consider an input sequence* $\mathbf{X}$ *from* $DSET^{(k)}$ *and corresponding TPG* $\mathcal{G}^{(k)}$. *We define* $\widehat{\mathcal{G}}^{(k)}(\mathbf{X}) \in \mathcal{G}^{(k)}$ *as the* highest probability SCC *for* $\mathbf{X}$ *in* $\mathcal{G}^{(k)}$ *such that* $\forall \mathbf{x} \in \widehat{\mathcal{G}}^{(k)}(\mathbf{X})$ *we have* $[\mathbb{S}(\mathbf{C}f_{\tilde{\mathbf{W}}}(\mathbf{X}))]_x = \|\mathbb{S}(\mathbf{C}f_{\tilde{\mathbf{W}}}(\mathbf{X}))\|_\infty$.

## 3 Defining topics

In order to answer our research questions regarding the dynamics of topic changes we need to define the concept of a topic. In the previous settings, a dataset DSET generates TPGs $\{\mathcal{G}^{(k)}\}_{k=1}^K$, but, conversely, an existing set of TPGs can generate DSET. Therefore, inspired by Ameisen et al. [1] that introduces attribution graphs to reveal the LLMs' internal computational structure, we define a topic as a set of TPGs:

**Definition 2** (*topic*). *A topic* $\mathbb{T}$ *is a set of TPGs* $\{\mathcal{G}^{(k)}\}_{k=1}^K$. *Given topic* $\mathbb{T}$ *defined by TPGs* $\{\mathcal{G}^{(k)}\}_{k=1}^K$, *input sequence* $\mathbf{X}$ *belongs to* $\mathbb{T}$ *if* $\forall \mathbf{x} \in \mathbf{X}, x \in \mathcal{G}^{(\bar{x})}$. *A sequence* $(\mathbf{X}, y)$ *is within* $\mathbb{T}$ *if* $\mathbf{X}$ *belongs to* $\mathbb{T}$ *and* $\forall \mathbf{x} \in \mathbf{X}, (y \Rightarrow x) \in \mathcal{G}^{(\bar{x})}$.

Our graph-based formulation aligns with recent advances in structured representations of LLMs [42, 52]. Given the finite number of edges, a DSET can be generated from $\mathbb{T}$ such that it can reconstruct the exact TPGs $\{\mathcal{G}^{(k)}\}_{k=1}^K$ that define $\mathbb{T}$, following the construction method in Li et al. [25]. This leads to the following reasonable assumption:

**Assumption 4.** *A DSET generated from any topic* $\mathbb{T}$ *defined by* $\{\mathcal{G}^{(k)}\}_{k=1}^K$ *exactly reconstructs back the TPGs* $\{\mathcal{G}^{(k)}\}_{k=1}^K$.

Detailed explanation is provided in Appendix B. This assumption enables the application of the results from Li et al. [25], with the concepts of topics and TPGs being used interchangeably.

**Definition 3** (*topic continuity*). *Given an input sequence* $\mathbf{X}$ *that belongs to* $\mathbb{T}$, *a weight matrix* $\mathbf{W}$ *is said to* keep *topic* $\mathbb{T}$ *for the input sequence* $\mathbf{X}$ *if* $\hat{y}_{\mathbf{W}} \in \widehat{\mathcal{G}}^{(k)}(\mathbf{X})$.

**Remark.** Given two topics, $\mathbb{T}_a$ and $\mathbb{T}_b$, with corresponding datasets $DSET_a$ and $DSET_b$, the union of $\{\mathcal{G}_a^{(k)}\}_{k=1}^K$ and $\{\mathcal{G}_b^{(k)}\}_{k=1}^K$ forms the TPGs for the mixed topics $\mathbb{T}_{ab}$, denoted by $\{\mathcal{G}_{ab}^{(k)}\}_{k=1}^K$.

It is clear that $\tilde{\mathbf{W}}_a$ trained only with $DSET_a$ will always *keep* topic $\mathbb{T}_a$.[3] But we could also obtain $\tilde{\mathbf{W}}_{ab}$ with a dataset combining $DSET_a$ and $DSET_b$ as training sets. The central question is whether $\tilde{\mathbf{W}}_{ab}$ *keeps* topic $\mathbb{T}_a$, given an input sequence $\mathbf{X}$ that belongs to $\mathbb{T}_a$, or if it instead predicts tokens that prompt a topic change.

## 4 Attention within mixed topics

Let's first understand how attention models assign priority to tokens within mixed-topic setting. For simplicity, we elaborate our results using a two-topic scenario, but it is straightforward to extend the results on multiple topics. Notice the self-attention embedding output is a linear combination of $\mathbf{X}$ given by $\mathbb{S}(\mathbf{X}\mathbf{W}\bar{\mathbf{x}})$. The embeddings in $\mathbf{X}$ corresponding to the highest entries in $\mathbb{S}(\mathbf{X}\mathbf{W}\bar{\mathbf{x}})$ will receive higher priority to predict the next token, therefore we can hypothesize that models in which $\mathbb{S}(\mathbf{X}\mathbf{W}\bar{\mathbf{x}})$ are ordered in a similar way will predict similar next tokens. This idea leads to our first main result which considers this situation within a mixed-topic setting:

**Theorem 2.** *Consider datasets* $DSET_a$ *and* $DSET_b$ *from topics* $\mathbb{T}_a$ *and* $\mathbb{T}_b$, *respectively. Let* $DSET_{ab}$ *be the union of* $DSET_a$ *and* $DSET_b$. *Suppose Assumptions 1, 2, 3 and 4 hold. Set loss function as* $\ell(u) = -\log(u)$. *Starting Algo-GD from any initial point with constant size* $\eta$ *and if* $\mathbf{W}_a^{svm} \neq \mathbf{0}$ *and* $\mathbf{W}_{ab}^{svm} \neq \mathbf{0}$; *for a given sequence* $\mathbf{X}$ *that belongs to* $\mathbb{T}_a$, *we have that* $\tilde{\mathbf{W}}_{ab}$ *preserves the attention priority of* $\mathbb{T}_a$ *on input* $\mathbf{X}$. *This is* $\forall i, j \in [T]$:

---

[3]*Notation:* The subscripts of weights and objects correspond to the associated topic. For instance $\tilde{\mathbf{W}}_a$ denotes the weights defined in Equation 2, obtained from $DSET_a$, which pertains to topic $\mathbb{T}_a$.

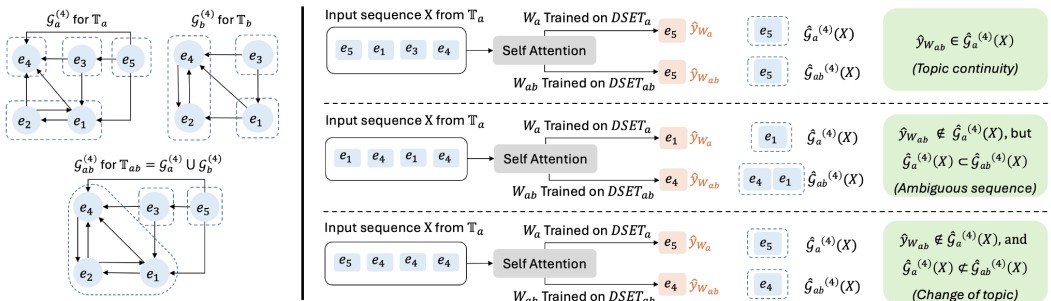

Figure 3: Depiction of each scenario in next token prediction. **Left:** Taking the last token $\mathbf{e}_4$ as an example, $\mathcal{G}_{ab}^{(4)}$ for $\mathbb{T}_{ab}$ is formed by the union of $\mathcal{G}_a^{(4)}$ and $\mathcal{G}_b^{(4)}$. The direction of edge is from output to input and the dotted square denotes the strongly-connected components (SCC) in which tokens have equal priority. **Right:** For each input sequence belonging to $\mathbb{T}_a$, we use a self-attention model trained on $\text{DSET}_a$ and another model trained on the mixed-topic dataset $\text{DSET}_{ab}$ to predict the next tokens, denoted as $\hat{y}_{\mathbf{w}_a}$ and $\hat{y}_{\mathbf{w}_{ab}}$, respectively. $\widehat{\mathcal{G}}_{ab}^{(4)}$ and $\widehat{\mathcal{G}}_a^{(4)}$ represent the *highest probability SCCs* (Definition 1) in mixed-topic setting and in $\mathbb{T}_a$, respectively. There are three scenarios, *topic continuity* (Definition 3), *ambiguous sequence* (Definition 4), and *change of topic* (Definition 5). The numeric details for each scenario are provided in Appendix C.5.

- *if* $\quad [\mathbb{S}(\mathbf{X}\tilde{\mathbf{W}}_a\bar{\mathbf{x}})]_i = [\mathbb{S}(\mathbf{X}\tilde{\mathbf{W}}_a\bar{\mathbf{x}})]_j$, *then* $\quad [\mathbb{S}(\mathbf{X}\tilde{\mathbf{W}}_{ab}\bar{\mathbf{x}})]_i = [\mathbb{S}(\mathbf{X}\tilde{\mathbf{W}}_{ab}\bar{\mathbf{x}})]_j$

- *if* $\quad [\mathbb{S}(\mathbf{X}\tilde{\mathbf{W}}_a\bar{\mathbf{x}})]_i > [\mathbb{S}(\mathbf{X}\tilde{\mathbf{W}}_a\bar{\mathbf{x}})]_j$, *then* $\quad [\mathbb{S}(\mathbf{X}\tilde{\mathbf{W}}_{ab}\bar{\mathbf{x}})]_i \geq [\mathbb{S}(\mathbf{X}\tilde{\mathbf{W}}_{ab}\bar{\mathbf{x}})]_j$

- *if* $\quad [\mathbb{S}(\mathbf{X}\tilde{\mathbf{W}}_a\bar{\mathbf{x}})]_i < [\mathbb{S}(\mathbf{X}\tilde{\mathbf{W}}_a\bar{\mathbf{x}})]_j$, *then* $\quad [\mathbb{S}(\mathbf{X}\tilde{\mathbf{W}}_{ab}\bar{\mathbf{x}})]_i \leq [\mathbb{S}(\mathbf{X}\tilde{\mathbf{W}}_{ab}\bar{\mathbf{x}})]_j$

This implies that for an input sequence $\mathbf{X}$, a model trained in a mixed-topic setting will maintain the priority of the topic to which $\mathbf{X}$ belongs. Consequently, the attention will be allocated in the same order as if the model had been trained exclusively on the original topic of $\mathbf{X}$. For the first input sequence $\mathbf{X} = [\mathbf{e}_5, \mathbf{e}_1, \mathbf{e}_3, \mathbf{e}_4]^\top$ from $\mathbb{T}_a$, as shown in Figure 3 (right), the predicted next token $\hat{y}_{\mathbf{w}_{ab}}$ is $\mathbf{e}_5$ and the *highest probability SCC* in mixed topics is $\widehat{\mathcal{G}}_{ab}^{(4)}(\mathbf{X}) = \{\mathbf{e}_5\}$. Since $\hat{y}_{\mathbf{w}_{ab}}$ belongs to $\widehat{\mathcal{G}}_{ab}^{(4)}(\mathbf{X})$, $\mathbf{W}_{ab}$ for input sequence $\mathbf{X}$ is considered as *topic continuity*, based on the Definition 3.

The only assumption about $\mathbf{X}$ on Theorem 2 is that it belongs to $\mathbb{T}_a$. However, if $\mathbf{X}$ belongs to $\mathbb{T}_a$ and $\mathbb{T}_b$, the priority will be preserved within both topics. Additionally, strict equality in the attention priority holds, but strict inequalities may not, as the union of their TPGs can form new SCCs. As illustrated on the left of Figure 3, $\mathcal{G}_a^{(4)}$ and $\mathcal{G}_b^{(4)}$ denote the TPGs corresponding to the last input token $\mathbf{e}_4$ for $\mathbb{T}_a$ and $\mathbb{T}_b$, respectively. In $\mathcal{G}_a^{(4)}$, the token priority is $\mathbf{e}_5 > \mathbf{e}_3 > \mathbf{e}_1 = \mathbf{e}_2 > \mathbf{e}_4$. In contrast, in $\mathcal{G}_{ab}^{(4)}$ for the mixed topics, the priority order is $\mathbf{e}_5 > \mathbf{e}_3 > \mathbf{e}_1 = \mathbf{e}_2 = \mathbf{e}_4$. The equality $\mathbf{e}_1 = \mathbf{e}_2$ from $\mathcal{G}_a^{(4)}$ is maintained in $\mathcal{G}_{ab}^{(4)}$, whereas the strict inequality $\mathbf{e}_2 > \mathbf{e}_4$ is relaxed to $\mathbf{e}_2 = \mathbf{e}_4$ in mixed topics, forming the new SCC, $\{\mathbf{e}_1, \mathbf{e}_2, \mathbf{e}_4\}$, in $\mathcal{G}_{ab}^{(4)}$.

## 5 Explaining topic shifts

The formation of new SCCs when combining datasets suggests that the highest priority SCC for some input sequences may increase in size in this new setting. This also suggests that topic shifts may arise from ambiguity within an input sequence rather than a straightforward change in topic. In our oracle analogy, gaining knowledge of both Topic A and Topic B might cause a conversation to be naturally followed within Topic A or also outside Topic A. We introduce the following definition to characterize this phenomenon:

**Definition 4** (*ambiguous sequence*). *Given $DSET_a$ and $DSET_b$ generated from two different topics $\mathbb{T}_a$ and $\mathbb{T}_b$. Denote $\mathbb{T}_{ab}$ as the combined topic defined by a combination of $DSET_a$ and $DSET_b$. A sequence $\mathbf{X}$ that belongs to $\mathbb{T}_a$ is* ambiguous *in $\mathbb{T}_{ab}$ with respect to $\mathbb{T}_a$ if $\tilde{\mathbf{W}}_{ab}$ does not keep topic $\mathbb{T}_a$ for $\mathbf{X}$, but $\widehat{\mathcal{G}}_a^{(\bar{x})}(\mathbf{X}) \subset \widehat{\mathcal{G}}_{ab}^{(\bar{x})}(\mathbf{X})$.*

Definition 4 defines an ambiguous sequence as one where the highest-probability next-token predictions include tokens from both within and outside the input topic, reflecting natural ambiguity from overlapping topics. Take the second input sequence $\mathbf{X} = [\mathbf{e}_1, \mathbf{e}_4, \mathbf{e}_1, \mathbf{e}_4]^\top$ in Figure 3 (right) as an example. $\widehat{\mathcal{G}}_a^{(4)}(\mathbf{X})$ is $\{\mathbf{e}_1\}$, as depicted in $\mathcal{G}_a^{(4)}$ from Figure 3 (left) and $\widehat{\mathcal{G}}_{ab}^{(4)}(\mathbf{X})$ is $\{\mathbf{e}_1, \mathbf{e}_4\}$, as shown in $\mathcal{G}_{ab}^{(4)}$ from Figure 3 (left). $\widehat{\mathcal{G}}_a^{(4)}(\mathbf{X})$ is a subset of $\widehat{\mathcal{G}}_{ab}^{(4)}(\mathbf{X})$, although $\hat{y}_{\mathbf{w}_{ab}} \notin \widehat{\mathcal{G}}_a^{(4)}(\mathbf{X})$. We can argue that the next token predicted from an ambiguous sequence cannot be considered as a topic change, as it lacks the clear trigger phenomenon observed in human cognition. To address this, we propose a formal definition for a topic change:

**Definition 5** (*change of topic*). *Given $DSET_a$ and $DSET_b$ generated from two topics $\mathbb{T}_a$ and $\mathbb{T}_b$, and a sequence $\mathbf{X}$ that belongs to $\mathbb{T}_a$. The weight matrix $\tilde{\mathbf{W}}_{ab}$ changes topic $\mathbb{T}_a$ for sequence $\mathbf{X}$ if $\tilde{\mathbf{W}}_{ab}$ does not keep topic $\mathbb{T}_a$ for $\mathbf{X}$ and $\mathbf{X}$ is not ambiguous in $\mathbb{T}_{ab}$ with respect to $\mathbb{T}_a$.*

In Figure 3 (right), $\mathbf{W}_{ab}$ changes topic for the last input sequence $\mathbf{X} = [\mathbf{e}_5, \mathbf{e}_4, \mathbf{e}_4, \mathbf{e}_4]^\top$, following the Definition 5. Building on the formal definitions of topic continuity, ambiguous sequences, and topic changes, we now present a necessary condition for a sequence to induce a topic change. This is achieved by introducing our final definition, grounded in the highest-priority SCC as determined by the order in the attention layer.

**Definition 6** (*highest priority SCC*). *Consider a sequence $\mathbf{X}$ that belongs to $\mathbb{T}$. We define $\dot{\mathcal{G}}^{(\bar{x})}(\mathbf{X}) \subseteq \mathcal{G}^{(\bar{x})}$ as the highest priority SCC for $\mathbf{X}$ in $\mathcal{G}^{(\bar{x})}$ such that $\forall x_i \in \dot{\mathcal{G}}^{(\bar{x})}(\mathbf{X})$ and $x_j \in \mathcal{G}^{(\bar{x})}$ we have $(x_i \Rightarrow x_j) \in \mathcal{G}^{(\bar{x})}$ or $(x_i \asymp x_j) \in \mathcal{G}^{(\bar{x})}$.*

**Theorem 3.** *Under the same settings and assumptions in Theorem 2, let $\mathbf{X}$ be a sequence that belongs to $\mathbb{T}_a$. If $\tilde{\mathbf{W}}_{ab}$ changes topic $\mathbb{T}_a$ for $\mathbf{X}$ then $\exists x_j \notin \dot{\mathcal{G}}_a^{(\bar{x})}(\mathbf{X})$ such that $\forall x_i \in \dot{\mathcal{G}}_a^{(\bar{x})}(\mathbf{X})$, the number of times $\mathbf{x}_j$ appears in $\mathbf{X}$ is greater than the number of times $\mathbf{x}_i$ appears in $\mathbf{X}$.*

Theorem 3 implies that, for a given sequence $\mathbf{X}$ from $\mathbb{T}_a$ and its corresponding TPG, a necessary condition for a topic change is the presence of a lower-priority token that appears more frequently than any of the higher-priority tokens. This can be intuitively understood through our analogy: if the oracle is following a conversation on Topic A but the conversation contains repeated components with lower importance in Topic A, its knowledge of Topic B may steer the response toward Topic B, thereby initiating a shift away from Topic A. A natural question arises: what do these findings imply in practice? Specifically, how does the probability of change of topic behave as the input sequence length or the topic ambiguity increases? The following theorem sheds light on these dynamics.

**Theorem 4.** *Under same settings and assumptions on datasets and training in Theorem 2, let $\mathbf{X}$ be a sequence that belongs to $\mathbb{T}_a$ with no repeated tokens, and $l$ be the number of elements in $\dot{\mathcal{G}}_a^{(\bar{x})}(\mathbf{X})$. Let $\mathbf{X}' = [\mathbf{x}_1' \, \mathbf{x}_2' \, \cdots \, \mathbf{x}_T']^\top$ be a random sequence of iid random tokens sampled from $\mathbf{X}$ such that for a fixed $p$, $p = \min_{x \in \dot{\mathcal{G}}_a^{(\bar{x})}(\mathbf{X})} \mathbb{P}(\mathbf{x}_i' = \mathbf{x})$. We have:*

1. *If $p > \max_{x \notin \dot{\mathcal{G}}_a^{(\bar{x})}(\mathbf{X})} \mathbb{P}(\mathbf{x}_i' = \mathbf{x})$, then $\lim_{T \to \infty} \mathbb{P}(\tilde{\mathbf{W}}_{ab}$ changes topic $\mathbb{T}_a$ for $\mathbf{X}') = 0$.*

2. *If $l$ increases then the probability that $\exists x_j' \notin \dot{\mathcal{G}}_a^{(\bar{x})}(\mathbf{X})$ such that $\forall x_i' \in \dot{\mathcal{G}}_a^{(\bar{x})}(\mathbf{X})$, $\mathbf{x}_j'$ outnumbers $\mathbf{x}_i'$ in $\mathbf{X}'$ does not increase.*

There are two implications of this theorem. First, as the input sequence length increases sufficiently, the likelihood of topic changes vanishes. Second, increasing $l$ raises the probability of overlap between topics and reduces the probability of satisfying the necessary conditions for a topic change, effectively creating a bound on the frequency of topic changes. In practice, consider the oracle analogy: if the oracle is following a sufficiently long conversation on a specific topic, it becomes exceedingly unlikely to shift topics. Similarly, as topics A and B become more interconnected, this increased ambiguity does not lead to more topic changes; rather, it may reduce their occurrence. This contrasts with human cognition, where longer conversations and greater inter-connectivity of knowledge increase the likelihood of spontaneous topic changes.

To illustrate Theorem 4 through simulations, we generate embeddings with $K = 10$ and $d = 16$. We approximate $\tilde{\mathbf{W}}_a$ and $\tilde{\mathbf{W}}_{ab}$ as the results obtained after $\tau = 8000$ iterations of Algo-GD. We quantify the proportion of test sequences in which $\tilde{\mathbf{W}}_{ab}$ keeps $\mathbb{T}_a$ (*keep topic*), proportion of ambiguous sequences in $\mathbb{T}_{ab}$ (*ambiguity*) and proportion in which $\tilde{\mathbf{W}}_{ab}$ changes topic (*change topic*). First, we

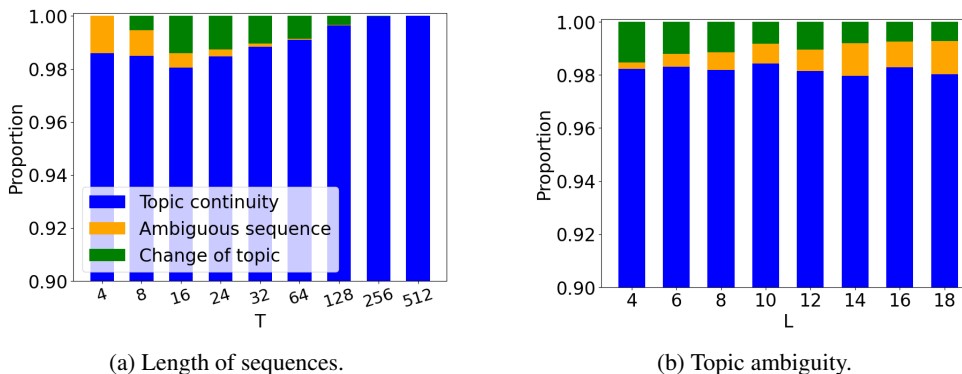

(a) Length of sequences.  (b) Topic ambiguity.

Figure 4: The proportion of topic continuity, ambiguous sequence, and change of topic as (a) input length and (b) topic ambiguity increase.

explore the effect of longer sequences by varying the length $T$ of the test sequences $\mathbf{Z}$. We increase $T$ from 4 to 512. Figure 4a illustrates how the proportion of *change topic* decreases as $T$ increases. Second, we investigate the effect of topic overlap with an increasing number of edges $L$. Intuitively, a higher $L$ results in an increase $l$ and a greater overlap between TPGs of different topics. We vary $L$ from 4 to 18. Figure 4b demonstrates that as $L$ increases, ambiguity increases, while the proportion of *change topic* doesn't increase. These two findings contrast with expectations derived from human cognition but align with the result of Theorem 4. Lastly, among the 85,000 test sequences generated for these experiments, 99.98% satisfy Theorem 3 (i.e., topic changes occur when a low-priority token appears more frequently than high-priority tokens). The remaining 0.02% mismatched cases are solely due to minor approximation discrepancies in the attention softmax. These results validate Theorem 3 (see simulation details in Appendix C).

## 6 Experiments in frontier LLMs

To prove Theorem 4 we work within the simplified, single-layer self-attention model of Li et al. [25]. Although this abstraction omits many hallmarks of contemporary LLMs (deep stacks of attention blocks, alternative cost functions, and other training heuristics), it offers a mathematically tractable setting that lets us derive interesting mathematical results. These results, in turn, can be used to understand how cutting-edge LLMs behave in terms of spontaneous topic changes. We empirically investigate such behavior on four frontier models: GPT-4o, Llama-3.3, Claude-3.7, and DeepSeek-V3.

**Real dataset.** We randomly select 100 arXiv papers published in March 2025 since the publicly disclosed knowledge cutoff dates for our study LLMs fall at the end of 2024 or earlier. This ensures that these models have not been trained on these data. We consider each paper as a different "topic".

**Experimental setup.** For two distinct papers A and B, and an input prompt ($\mathbf{X}$) from paper A, we consider a measure of *topic continuity* as the cosine similarity between the embeddings of the texts generated when the LLM has contextual knowledge solely from paper A ($\hat{y}_{\mathbf{W_a}}$) and when the LLM has contextual knowledge from both paper A and B ($\hat{y}_{\mathbf{W_{ab}}}$). We treat this cosine similarity as an empirical proxy for our formal definition of *topic continuity* (Definition 3): therefore the larger the similarity, the smaller the chance that the model has led to a *change of topic*. This proxy suggests two testable consequences which become the empirical counterpart of our Theorem 4: (1) *cosine similarity is expected to increase with the length of the input prompt*, and (2) *it is not expected to decrease with increasing ambiguity in paper A and paper B*.

To more closely align with our theoretical framework, where a model gains knowledge of topic A and incrementally gains knowledge of topic B, we implement a Retrieval-Augmented Generation (RAG) approach, retrieving information exclusively from paper A or jointly from papers A and B [51]. Based on the input prompt, we retrieve the top 3 most relevant excerpts from paper A or paper B to form the contextual knowledge set A or set B. The combined contextual knowledge set is simply the union of sets A and B. We add set A to the input prompt to obtain the generated text with sole knowledge of paper A ($\hat{y}_{\mathbf{W_a}}$), and we add the combined set to the input prompt to obtain the generated text with

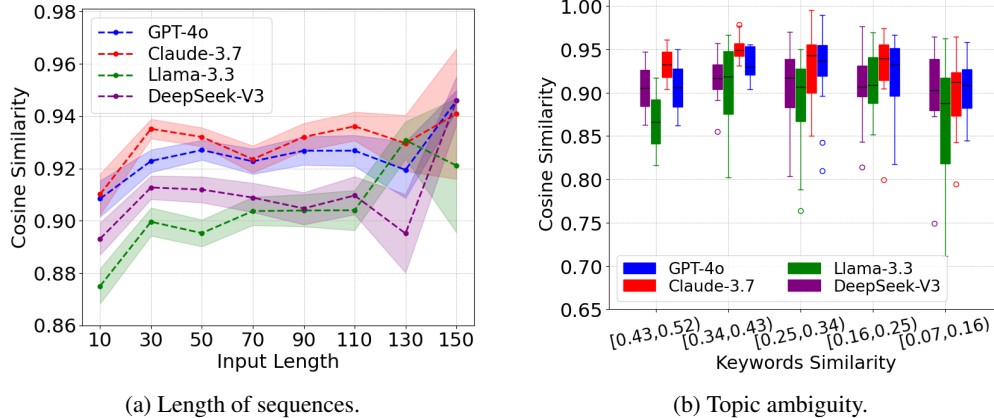

(a) Length of sequences.

(b) Topic ambiguity.

Figure 5: Similarity between continuations generated with single-topic and mixed-topic knowledge as (a) input length and (b) topic ambiguity increase.

combined knowledge of paper A and B ($\hat{y}_{\mathbf{W_{ab}}}$). To closely follow our greedy decoding approach in our theoretical framework, we set the temperature parameter to 0 for all LLMs.

We designate each paper as paper A and randomly select 5 different papers from the remaining 99 papers as distinct paper B. For each input segment, we calculate the average cosine similarity between $\hat{y}_{\mathbf{W_a}}$ and $\hat{y}_{\mathbf{W_{ab}}}$ across these five pairs of paper A and paper B, using each LLM. The results for each LLM are averaged over all 100 papers. See additional experimental details in Appendix D.

**Experiment 1: Impact of input length.** We use the first $10, 30, \dots, 150$ words from each paper A's abstract as the input prompt. Figure 5a shows, for each LLM, the average cosine similarity as a function of input length; shaded bands indicate 95% confidence intervals. Across all models, similarity tends to increase with input length, aligning with the behavior predicted by Theorem 4. Appendix D.3.1 presents an additional experiment in which we extend the input length to 1210 words extracted from each paper's introduction; the results further support our conclusions.

**Experiment 2: Impact of topic ambiguity.** We fix the input prompt length to the first $80$ words of each paper A's abstract. We quantify topic ambiguity by the average similarity among each paper A's keywords: lower keyword similarity signifies higher probability of overlap between paper A and other papers, consistent with our setup in Theorem 4. We partition the papers into five equal-width bins along this ambiguity spectrum. Figure 5b summarizes the results: each boxplot shows the distribution of cosine similarities within an ambiguity bin, with the x-axis ordered from least to most ambiguous. Across all LLMs the median similarity does not seem to decrease, in agreement with the prediction of Theorem 4. In Appendix D.3.2, we present an additional experiment using an alternative ambiguity measure based on cross-paper keyword similarity, yielding results consistent with our conclusions.

Taken together, the two experiments provide preliminary empirical support for Theorem 4, showing that its prediction, derived from a single-layer self-attention toy model, can be extended to today's deep, multi-layer LLMs. Crucially, an important divergence between machine and human cognition persists in these frontier models: neither longer prompts nor greater topic ambiguity appreciably increases the likelihood of a spontaneous topic change.

## 7 Related work

**Training and generalization of Transformer.** **(1) Properties of Softmax**. The self-attention mechanism employs the softmax function to selectively emphasize different parts of the input. Gu et al. [16], Goodfellow et al. [13], and Deng et al. [10] underscore the pivotal role of the softmax function in shaping attention distributions, influencing how models process and prioritize information within input sequences. Bombari and Mondelli [5] examined the word sensitivity of attention layers, revealing that softmax-based attention layers are adept at capturing the significance of individual words. However, recent work has also pointed out limitations of the softmax function [41, 10]. **(2) Optimization in attention-based models.** Additionally, recent research interprets Transformer

models as kernel machines, akin to support vector machines (SVMs), with self-attention layers performing maximum margin separation in the token space [48, 49, 25, 21]. **(3) Chain-of-Thought (CoT) and In-Context Learning (ICL).** Moreover, transformers exhibit remarkable abilities in generalization through ICL, where models effectively learn from contextual cues during inference [6, 56, 37]. CoT prompting [54, 57, 44, 24] enhances this by breaking down reasoning processes into intermediate steps, highlighting the emergent reasoning abilities of transformers. **(4) Improvement efficiency of transformers.** Recent advancements aim to improve the computational efficiency of transformers [22, 7, 47, 53], ensuring their viability for large-scale deployment while maintaining or enhancing their representational capabilities.

**Next token prediction in LLMs. (1) Theoretical and architectural innovations.** Shannon [43]'s foundational work laid the groundwork for estimating the predictability of natural language sequences, providing a basis for subsequent advances in language modeling. Recent studies have expanded our understanding of how LLMs anticipate future tokens from internal hidden states, offering valuable insights into the efficiency and effectiveness of Transformer-based architectures [17, 38, 45]. Despite their impressive predictive capabilities, these models face fundamental limitations. For instance, Bachmann and Nagarajan [3] highlights the shortcomings of teacher-forced training, emphasizing how this approach can fail and suggesting strategies to improve model robustness. **(2) Efficiency and Optimization.** Goyal et al. [14] introduces a novel method that incorporates a deliberate computation step before output generation, enhancing reasoning capabilities. Additionally, Gloeckle et al. [12] advocates for multi-token prediction, which significantly improves both efficiency and speed.

**Self-Attention and topic dynamics.** Advancements in self-attention research have deepened our understanding of how transformers handle evolving semantic contexts. Prior work has explored diverse aspects of topic modeling, such as dynamic topic structures [35], hierarchical relationships [29], topic-aware attention mechanisms [39], and the mechanistic underpinnings of topic representation [26]. While these studies provide insights into managing static and hierarchical topic structures, our work focuses on the topic changes with the given input sequences from a specific topic.

# 8 Discussion

Our theoretical analysis on self-attention models and empirical investigations on modern LLMs reveal fundamental clues regarding the distinctions between model-based spontaneous topic changes and spontaneous human thought, a phenomenon that is critical for comparing conversational dynamics across humans and AI. In an era of growing concern about AI's cognitive resemblance to humans, our framework provides preliminary results differentiating these phenomena, thereby opening pathways for future interdisciplinary research at the interface of artificial and human cognition.

**Limitations.** Our theoretical framework builds on the same simplified single-layer self-attention model with a log-loss objective from Li et al. [25] and defines topics as TPGs. These abstractions do not fully capture the complexities of contemporary LLMs, including deep attention architectures, alternative loss functions, and diverse training objectives. Despite loosening these assumptions, our experiments suggest that the essence of our core theoretical conclusions holds across modern LLMs within our framework of study. Future work will investigate how broadly these theoretical insights generalize to complex architectures, for example within longer context windows and/or LLM outputs.

**Code.** The source code can be found on GitHub: https://github.com/muminjia/Dynamics-of-Spontaneous-Topic-Changes

# Acknowledgments and Disclosure of Funding

This work was supported by the Natural Sciences and Engineering Research Council of Canada (NSERC) under grant DGECR-2022-04531. The authors thank Prof. Yingcong Li, the anonymous reviewers, and the session chair for their valuable feedback and insightful suggestions, which greatly improved the quality of this work.

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

# Supplementary Materials

# A Technical proofs

## A.1 Proof of Lemma 1

Let $\mathbf{a} = \mathbb{S}(\mathbf{X}\tilde{\mathbf{W}}\bar{\mathbf{x}})$.

$$\mathbf{C}f_{\tilde{\mathbf{W}}}(\mathbf{X}) = \mathbf{C}\left(\mathbf{X}^\top \mathbb{S}\left(\mathbf{X}\tilde{\mathbf{W}}\bar{\mathbf{x}}\right)\right) \tag{3}$$

$$= \mathbf{C}\left(\mathbf{X}^\top \mathbf{a}\right) \tag{4}$$

$$= \begin{bmatrix} \sum_{i=1}^{T} a_i\left(\mathbf{c}_1^\top \cdot \mathbf{x}_i\right) \\ \sum_{i=1}^{T} a_i\left(\mathbf{c}_2^\top \cdot \mathbf{x}_i\right) \\ \vdots \\ \sum_{i=1}^{T} a_i\left(\mathbf{c}_K^\top \cdot \mathbf{x}_i\right) \end{bmatrix}. \tag{5}$$

$$\tag{6}$$

Let $k_i$ be the number of times token $\mathbf{x}_i$ appears in $\mathbf{X}$. Then,

$$[\mathbf{C}f_{\tilde{\mathbf{W}}}(\mathbf{X})]_{x_i} = k_i a_i.$$

From Assumption 3 we have that

$$[\mathbf{C}f_{\tilde{\mathbf{W}}}(\mathbf{X})]_{x_i} = [\mathbf{C}f_{\tilde{\mathbf{W}}}(\mathbf{X})]_{x_j} \iff a_i = a_j \tag{7}$$

$$\iff (x_i \asymp x_j) \in \mathcal{G}^{(\bar{x})} \text{ or } x_i = x_j. \tag{8}$$

If $x_i \neq x_j$ then $x_i$ and $x_j$ are in the same SCC. $\qquad\square$

## A.2 Proof of Lemma 2

**Lemma 2.** *For an input sequence* $\mathbf{X}$ *that belongs to* $\mathbb{T}$ *and* $i, j \in [T]$,

- $[\mathbb{S}(\mathbf{X}\tilde{\mathbf{W}}\bar{\mathbf{x}})]_i = [\mathbb{S}(\mathbf{X}\tilde{\mathbf{W}}\bar{\mathbf{x}})]_j \iff (x_i \asymp x_j) \in \mathcal{G}^{(\bar{x})}$ *or* $i = j$.

- $[\mathbb{S}(\mathbf{X}\tilde{\mathbf{W}}\bar{\mathbf{x}})]_i < [\mathbb{S}(\mathbf{X}\tilde{\mathbf{W}}\bar{\mathbf{x}})]_j \iff (x_j \Rightarrow x_i) \in \mathcal{G}^{(\bar{x})}$.

- $[\mathbb{S}(\mathbf{X}\tilde{\mathbf{W}}\bar{\mathbf{x}})]_i > [\mathbb{S}(\mathbf{X}\tilde{\mathbf{W}}\bar{\mathbf{x}})]_j \iff (x_i \Rightarrow x_j) \in \mathcal{G}^{(\bar{x})}$.

*Proof.* Since $\mathbf{X}$ belongs to $\mathbb{T}$, $\forall \mathbf{x} \in \mathbf{X}$ we have $x \in \mathcal{G}^{(\bar{x})}$, therefore from the construction of TPGs by Li et al. [25], for every $\mathbf{x}_i, \mathbf{x}_j \in \mathbf{X}$ we have one of the these relationships: $(x_i \Rightarrow x_j)$, $(x_j \Rightarrow x_i)$, $(x_i \asymp x_j)$ or $x_i = x_j$. From the constraints in Algo-GD:

- $[\mathbb{S}(\mathbf{X}\tilde{\mathbf{W}}\bar{\mathbf{x}})]_i = [\mathbb{S}(\mathbf{X}\tilde{\mathbf{W}}\bar{\mathbf{x}})]_j \iff (\mathbf{x}_i - \mathbf{x}_j)^\top \tilde{\mathbf{W}}\bar{x} = 0 \iff (x_j \asymp x_i) \in \mathcal{G}^{(\bar{x})}$ or $i = j$.

- $[\mathbb{S}(\mathbf{X}\tilde{\mathbf{W}}\bar{\mathbf{x}})]_i > [\mathbb{S}(\mathbf{X}\tilde{\mathbf{W}}\bar{\mathbf{x}})]_j \iff (\mathbf{x}_i - \mathbf{x}_j)^\top \tilde{\mathbf{W}}\bar{x} > 1 \iff (x_j \Rightarrow x_i) \in \mathcal{G}^{(\bar{x})}$.

- $[\mathbb{S}(\mathbf{X}\tilde{\mathbf{W}}\bar{\mathbf{x}})]_i < [\mathbb{S}(\mathbf{X}\tilde{\mathbf{W}}\bar{\mathbf{x}})]_j \iff (\mathbf{x}_i - \mathbf{x}_j)^\top \tilde{\mathbf{W}}\bar{x} < 1 \iff (x_i \Rightarrow x_j) \in \mathcal{G}^{(\bar{x})}$.

$\qquad\square$

## A.3 Proof of Lemma 3

**Lemma 3.** *For an input sequence* $\mathbf{X}$ *that belongs to* $\mathbb{T}$,

$$\dot{\mathcal{G}}^{(\bar{x})}(\mathbf{X}) = \left\{ x_i \mid [\mathbb{S}(\mathbf{X}\tilde{\mathbf{W}}\bar{\mathbf{x}})]_i = \|\mathbb{S}(\mathbf{X}\tilde{\mathbf{W}}\bar{\mathbf{x}})\|_\infty \right\}.$$

*Proof.* Let $G = \{x_i \mid [\mathbb{S}(\mathbf{X}\tilde{\mathbf{W}}\bar{\mathbf{x}})]_i = \|\mathbb{S}(\mathbf{X}\tilde{\mathbf{W}}\bar{\mathbf{x}})\|_\infty\}$. From Lemma 2 $\forall x_i, x_j \in G$, $(x_i \asymp x_j) \in \mathcal{G}^{(\bar{x})}$. Therefore all elements in $G$ belong to the same SCC. Also from Lemma 2, $\forall x_i \in G, x_j \notin G$ we have $(x_i \Rightarrow x_j) \in \mathcal{G}^{(\bar{x})}$. This means that every element in $G$ has the highest priority among tokens in $\mathbf{X}$ concluding our proof. $\qquad\square$

## A.4 Proof of Theorem 2

From construction, $\forall k \in [K]$, $\mathcal{G}_a^{(k)} \subseteq \mathcal{G}_{ab}^{(k)}$. This means that $\forall \mathbf{x}_i, \mathbf{x}_j \in \mathbf{X}$, we have:

- if $(x_i \asymp x_j) \in \mathcal{G}_a^{(\bar{x})}$ then $(x_i \asymp x_j) \in \mathcal{G}_{ab}^{(\bar{x})}$
- if $(x_j \Rightarrow x_i) \in \mathcal{G}_a^{(\bar{x})}$ then $(x_j \Rightarrow x_i) \in \mathcal{G}_{ab}^{(\bar{x})}$ or $(x_i \asymp x_j) \in \mathcal{G}_{ab}^{(\bar{x})}$
- if $(x_i \Rightarrow x_j) \in \mathcal{G}_a^{(\bar{x})}$ then $(x_i \Rightarrow x_j) \in \mathcal{G}_{ab}^{(\bar{x})}$ or $(x_i \asymp x_j) \in \mathcal{G}_{ab}^{(\bar{x})}$

Combining with Lemma 2:

- $[\mathbb{S}(\mathbf{X}\tilde{\mathbf{W}}_a\bar{\mathbf{x}})]_i = [\mathbb{S}(\mathbf{X}\tilde{\mathbf{W}}_a\bar{\mathbf{x}})]_j \iff (x_i \asymp x_j) \in \mathcal{G}_a^{(\bar{x})}$ or $i = j$, then $(x_i \asymp x_j) \in \mathcal{G}_{ab}^{(\bar{x})}$ or $i = j \iff [\mathbb{S}(\mathbf{X}\tilde{\mathbf{W}}_{ab}\bar{\mathbf{x}})]_i = [\mathbb{S}(\mathbf{X}\tilde{\mathbf{W}}_{ab}\bar{\mathbf{x}})]_j$

- $[\mathbb{S}(\mathbf{X}\tilde{\mathbf{W}}_a\bar{\mathbf{x}})]_i < [\mathbb{S}(\mathbf{X}\tilde{\mathbf{W}}_a\bar{\mathbf{x}})]_j \iff (x_j \Rightarrow x_i) \in \mathcal{G}_a^{(\bar{x})}$ then $(x_j \Rightarrow x_i) \in \mathcal{G}_{ab}^{(\bar{x})}$ or $(x_i \asymp x_j) \in \mathcal{G}_{ab}^{(\bar{x})} \iff [\mathbb{S}(\mathbf{X}\tilde{\mathbf{W}}_{ab}\bar{\mathbf{x}})]_i \leq [\mathbb{S}(\mathbf{X}\tilde{\mathbf{W}}_{ab}\bar{\mathbf{x}})]_j$

- $[\mathbb{S}(\mathbf{X}\tilde{\mathbf{W}}_a\bar{\mathbf{x}})]_i > [\mathbb{S}(\mathbf{X}\tilde{\mathbf{W}}_a\bar{\mathbf{x}})]_j \iff (x_i \Rightarrow x_j) \in \mathcal{G}_a^{(\bar{x})}$ then $(x_i \Rightarrow x_j) \in \mathcal{G}_{ab}^{(\bar{x})}$ or $(x_i \asymp x_j) \in \mathcal{G}_{ab}^{(\bar{x})} \iff [\mathbb{S}(\mathbf{X}\tilde{\mathbf{W}}_{ab}\bar{\mathbf{x}})]_i \geq [\mathbb{S}(\mathbf{X}\tilde{\mathbf{W}}_{ab}\bar{\mathbf{x}})]_j$ $\qquad\square$

## A.5 Proof of Theorem 3

Let $\mathbf{a} = \mathbb{S}(\mathbf{X}\tilde{\mathbf{W}}_a\bar{\mathbf{x}})$ and $\mathbf{b} = \mathbb{S}(\mathbf{X}\tilde{\mathbf{W}}_{ab}\bar{\mathbf{x}})$. Without loss of generality, suppose $\mathbf{a}$ is in decreasing order $a_1 \geq \cdots \geq a_T$. From Theorem 2, we also have $b_1 \geq \cdots \geq b_T$. Let $k_i$ be the number of times token $\mathbf{x}_i$ appears in $\mathbf{X}$. Following an analogous procedure as in Lemma 1 we get

$$\left[\mathbf{C}f_{\tilde{\mathbf{W}}_a(\tau)}(\mathbf{X})\right]_{x_i} = k_i a_i \tag{9}$$

$$\left[\mathbf{C}f_{\tilde{\mathbf{W}}_{ab}(\tau)}(\mathbf{X})\right]_{x_i} = k_i b_i \tag{10}$$

We will proof the contrapositive: If $\exists x_i \in \dot{\mathcal{G}}_a^{(\bar{x})}(\mathbf{X})$ such that $k_i \geq k_j$ for all $j \in [K]$, then there is no change of topic, so $\tilde{\mathbf{W}}_{ab}$ *keeps* topic $\mathbb{T}_a$ for input sequence $\mathbf{X}$, or $\mathbf{X}$ is *ambiguous* in $\mathbb{T}_{ab}$ with respect to $\mathbb{T}_a$.

From Lemma 3, if $x_i \in \dot{\mathcal{G}}_a^{(\bar{x})}(\mathbf{X})$, we have $a_i \geq a_j$ for all $j \in [K]$. Suppose $\exists x_i \in \dot{\mathcal{G}}_a^{(\bar{x})}(\mathbf{X})$ such that $k_i \geq k_j$ for all $j \in [K]$, we have that $k_i a_i \geq k_j a_j$ for all $j \in [K]$ then $x_i \in \widehat{\mathcal{G}}_a^{(\bar{x})}(\mathbf{X})$. Analogously since $b_i \geq b_j$, $x_i \in \widehat{\mathcal{G}}_{ab}^{(\bar{x})}(\mathbf{X})$. If $\exists x_l \in \widehat{\mathcal{G}}_a^{(\bar{x})}(\mathbf{X})$ with $x_l \neq x_i$ then $k_l a_l \geq k_j a_j$ for all $j \in [K]$, then $k_l a_l = k_i a_i$. Therefore from Assumption 3 and Lemma 3, $(x_l \asymp x_i) \in \mathcal{G}_a^{(\bar{x})}$. Analogously $(x_l \asymp x_i) \in \mathcal{G}_{ab}^{(\bar{x})}$. This means that if $\exists x_i \in \dot{\mathcal{G}}_a^{(\bar{x})}(\mathbf{X})$ such that $k_i \geq k_j$ for all $j \in [K]$, then $\widehat{\mathcal{G}}_a^{(\bar{x})}(\mathbf{X}) \subseteq \widehat{\mathcal{G}}_{ab}^{(\bar{x})}(\mathbf{X})$. Then $\tilde{\mathbf{W}}_{ab}$ *keeps* topic $\mathbb{T}_a$ for input sequence $\mathbf{X}$, or $\mathbf{X}$ is *ambiguous* in $\mathbb{T}_{ab}$ with respect to $\mathbb{T}_a$. $\qquad\square$

## A.6 Proof of Theorem 4

1. This is a direct consequence from the law of large numbers. If $T \to \infty$ the proportion of each token will match the probability. Since $p > \max_{\mathbf{x} \notin \dot{\mathcal{G}}_a^{(\bar{x})}(\mathbf{X})} \mathbb{P}(\mathbf{x}'_i = \mathbf{x})$, then the probability that $\exists x'_j \notin \dot{\mathcal{G}}_a^{(\bar{x})}(\mathbf{X})$ such that $\forall x'_i \in \dot{\mathcal{G}}_a^{(\bar{x})}(\mathbf{X})$, the number of times $\mathbf{x}'_j$ appears in $\mathbf{X}'$ is greater than the number of times $\mathbf{x}'_i$ appears in $\mathbf{X}'$ will go to zero, and therefore the probability of change topics will do it also.

2. Without loss of generality suppose $\dot{\mathcal{G}}_a^{(\bar{x})}(\mathbf{X}) = \{x_1, x_2, \cdots, x_l\}$. Clearly if we prove the result assuming $\forall x \in \dot{\mathcal{G}}_a^{(\bar{x})}(\mathbf{X})$, $p = \mathbb{P}(\mathbf{x}'_i = \mathbf{x})$, we will also have it for the more general case $p = \min_{\mathbf{x} \in \dot{\mathcal{G}}_a^{(\bar{x})}(\mathbf{X})} \mathbb{P}(\mathbf{x}'_i = \mathbf{x})$.

   Let $\mathbf{X}'_l = [\mathbf{x}'_{1,l}\ \mathbf{x}'_{2,l}\ \cdots\ \mathbf{x}'_{T,l}]^\top$ be a random sequences generated as described in the theorem, where the size of $\dot{\mathcal{G}}_a^{(\bar{x})}(\mathbf{X})$ is $l$. Let $k_{i,l}$ be the number of times $\mathbf{x}_i$ is selected in $\mathbf{X}'_l$. Let

$A_l = \max_{1 \le i \le l} k_{i,l}$ and $B_l = \max_{l+1 \le i \le K} k_{i,l}$. Let $P(l) = \mathbb{P}(B_l > A_l)$. We want to prove $P(l+1) \le P(l)$. We construct a coupling between $\mathbf{X}'_l$ and $\mathbf{X}'_{l+1}$ by performing $T$ independent trials. For each trial $i$ we generate a uniform random variable $U_i$ in $[0, 1]$ and we choose tokens in $\mathbf{X}'_l$ and $\mathbf{X}'_{l+1}$ in this way:

- If $U_i \le pl$ both the selected tokens $x'_{i,l}$ and $x'_{i,l+1}$ are in $\{x_1, x_2, \cdots, x_l\}$.
- If $pl < U_i \le p(l+1)$, we select $x'_{i,l} = x_{l+1}$ if $U_i \le pl + q$ or $x'_{i,l} = x_{l+2}$ otherwise, and we select $x'_{i,l+1} = x_{l+1}$; where $q$ is the probability of choosing $x_{l+1}$ in $\mathbf{X}'_l$. Since $p > q$, there is an interval where $x_{i,l} = x_{l+2}$ but $x_{i,l+1} = x_{l+1}$.
- If $U_i > p(l+1)$, then both the selected tokens $x'_{i,l}$ and $x'_{i,l+1}$ are in $\{x_{l+2}, x_2, \cdots, x_l\}$. Notice that the probability of choosing $x_i$ in $\mathbf{X}'_{l+1}$ for $i \ge l+2$ decreases because $p$ is constant.

From the previous coupling we have that $k_{i,l} = k_{i,l+1}$ for $1 \le i \le l$, $k_{l+1,l} \le k_{l+1,l+1}$ for $i = l+1$, and $k_{i,l} \ge k_{i,l+1}$ for $i \ge l+2$. This means that $A_{l+1} = \max(A_l, k_{l+1,l+1}) \ge A_l$ and $B_{l+1} = \max_{l+2 \le i \le K} k_{i,l+1} \le B_l$. Therefore $P(l+1) = \mathbb{P}(B_{l+1} > A_{l+1}) \le \mathbb{P}(B_l > A_l) = P(l)$.

$\square$

# B   Explanation of Assumption 4

As illustrated in Figure 6, the dataset for $\mathbb{T}_a$ and the dataset for $\mathbb{T}_b$ demonstrate interchangeability with $\mathcal{G}_a^{(4)}$ and $\mathcal{G}_b^{(4)}$, respectively.

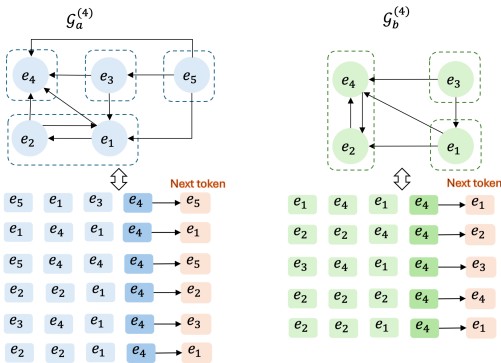

Figure 6: Illustration of Assumption 4. Here are two datasets related to the TPGs, $\mathcal{G}_a^{(4)}$ and $\mathcal{G}_b^{(4)}$, from Figure 3 (left). From the directed arrows in $\mathcal{G}_a^{(4)}$, we can generate a dataset with the last token $\mathbf{e_4}$ for $\mathbb{T}_a$, which can reconstruct back the $\mathcal{G}_a^{(4)}$. A similar process applies for the $\mathcal{G}_b^{(4)}$.

# C   Detailed simulation studies with single-layer self-attention

## C.1   Simulation process

**Theoretical TPGs generation.** For each token $e_k$, $L$ edges are randomly selected to construct the theoretical TPG $\mathcal{G}_{theor}^{(k)}$ for $e_k$, ensuring that $e_k$ is involved, as either a source or destination node. Based on these selected edges, we add additional edges from $e_k$ to all other tokens included in $L$ edges, thereby ensuring that all tokens in $\mathcal{G}_{theor}^k$ can be reached by $e_k$. Thus, we obtain the theoretical TPGs $\{\mathcal{G}_{a,theor}^{(k)}\}_{k=1}^K$ for Topic A . This process is repeated to generate another group of theoretical TPGs $\{\mathcal{G}_{b,theor}^{(k)}\}_{k=1}^K$ for the Topic B. Let $\mathcal{G}_{a,theor}^{(k)}$ and $\mathcal{G}_{b,theor}^{(k)}$ combine for each $k$, we obtain the theoretical TPGs for topics combinations $\{\mathcal{G}_{ab,theor}^{(k)}\}_{k=1}^K$.

**Training Dataset Generation.** Generate training datasets $\text{DSET}_a$ and $\text{DSET}_b$ based on $\{\mathcal{G}_{a,theor}^{(k)}\}_{k=1}^K$ and $\{\mathcal{G}_{b,theor}^{(k)}\}_{k=1}^K$, respectively. For each input sequence in DSET, the sequence length $T_{train}$ is 4, which means $\mathbf{X} = [\mathbf{x}_1 \, \mathbf{x}_2 \, \cdots \, \mathbf{x}_{T_{train}}]^\top \in \mathbb{R}^{T_{train} \times d}$ with $\mathbf{x}_i$ from $\mathbf{E} = [\mathbf{e}_1, \mathbf{e}_2, ... \mathbf{e}_K]^\top$. $\mathbf{e}_k$ is randomly selected as the last token and other tokens (other input tokens and the next predicted token) are chosen based on $\mathcal{G}_{theor}^{(k)}$. Specifically, the next token $\mathbf{e}_{T_{train}+1}$ is determined by sampling with the weighted probability in $\mathcal{G}_{theor}^{(k)}$, where the weight for each token corresponds to the number of outcoming edges. Given Assumption 2, we randomly choose the position of the next token in the input sequence. Then, the remaining input tokens are randomly selected from tokens connected by incoming edges from $\mathbf{e}_k$ (i.e., $\mathbf{e}_k \to \mathbf{e}_i$) and placed in the random position within the input sequence. This process is repeated $n$ times to generate training data for each topic respectively. Empirical TPGs $\{\mathcal{G}_{a,empir}^{(k)}\}_{k=1}^K$ and $\{\mathcal{G}_{b,empir}^{(k)}\}_{k=1}^K$ are derived from the training datasets $\text{DSET}_a$ and $\text{DSET}_b$. According to Assumption 4, the empirical TPGs $\{\mathcal{G}_{empir}^{(k)}\}_{k=1}^K$ are expected to be identical to the theoretical TPGs $\{\mathcal{G}_{theor}^{(k)}\}_{k=1}^K$ for each topic. The experiments are conducted with 5000 instances, with each parameter setting evaluated over 50 epochs, consisting of 100 sequences per epoch.

**Trained attention weights.** We employ a single-layer attention mechanism implemented in PyTorch. The model is trained using the SGD optimizer with a learning rate $\eta = 0.01$ for 8000 iterations. The training of attention weights is divided into two stages for each instance: (1) computing $\mathbf{W}^{\text{svm}}$ for each topic;[4] (2) get $\mathbf{W}(\tau)$ at each iteration for each topic. In Stage (1), prior to using the CVXPY package to get $\mathbf{W}^{\text{svm}}$, SCCs are identified for each TPG derived from the using *Tarjan's algorithm*. Afterward, $\mathbf{W}^{\text{svm}}$ is normalized to ensure consistency in subsequence computations. In Stage (2), the $MLayerAttn$ function encapsulates the architecture of a single-layer attention-based model. The training function is then used to optimize the attention weights by minimizing the loss defined in ERM. Finally, the correlation between $\mathbf{W}^{\text{svm}}$ and $\mathbf{W}(\tau)$ is calculated using the dot product.

**Next token prediction.** To differentiate the input sequence length of the testing data from that of the training data, we introduce $T_{test}$. TPGs based on the training dataset $\text{DSET}_a$ are utilized to generate test datasets consisting of 100 sequences from $\mathbb{T}_a$ per epoch. Specifically, the last token $\mathbf{x}_{T_{test}}$ of the test input sequence is randomly selected from $K$ tokens (i.e. $\mathbf{x}_{T_{test}} = \mathbf{e}_k$) and the remaining input tokens are randomly chosen based on the SCCs of $\mathcal{G}_a^k$, where tokens with higher priority are assigned greater weights. For instance, in $\mathcal{G}_a^4$, tokens $\mathbf{e}_1, \mathbf{e}_2, \mathbf{e}_4$ are captured with the priority order $\mathbf{e}_1 = \mathbf{e}_2 > \mathbf{e}_4$. The weights assigned to input tokens $\mathbf{e}_1, \mathbf{e}_2$, and $\mathbf{e}_4$ are $0.4, 0.4$, and $0.2$, respectively. It reflects that $\mathbf{e}_1$ and $\mathbf{e}_2$ are in the same higher-priority SCC, thus having greater weights compared to $\mathbf{e}_4$. Intuitively, tokens within the same SCC are more likely to co-occur than those from different SCCs. This approach enables the generated test input sequences to mimic real word relationships and reflect their contextual groupings. Following the generation of the test dataset from $\mathbb{T}_a$, the next tokens $\hat{y}_{\mathbf{w_a}}$ and $\hat{y}_{\mathbf{w_{ab}}}$ are predicted by Equation 1, with $\mathbf{W}_a$ and $\mathbf{W}_{ab}$ obtained from the last iteration. To reduce the potential numerical issues in the outputs, $\mathbb{S}(\mathbf{X}\mathbf{W}\bar{\mathbf{x}})$ is rounded to three decimals, ensuring that tokens within the same SCC yield consistent softmax outputs.

## C.2 Additional experiments to support Theorem 2

To further illustrate Theorem 2 we define the *attention priority similarity* of weights $\mathbf{W}'$ relative to $\mathbf{W}$ for a sequence $\mathbf{X}$ as: $R_{\mathbf{W},\mathbf{W}'}(\mathbf{X}) =$

$$\frac{1}{T-1} \sum_{j=1}^{T-1} g\left( [\mathbb{S}(\mathbf{X}\mathbf{W}'\bar{\mathbf{x}})]_{i_j} - [\mathbb{S}(\mathbf{X}\mathbf{W}'\bar{\mathbf{x}})]_{i_{j+1}} \right),$$

where $i_1, \cdots, i_T$ is a permutation of $1, \cdots, T$ such that $[\mathbb{S}(\mathbf{X}\mathbf{W}\bar{\mathbf{x}})]_{i_1} \geq \cdots \geq [\mathbb{S}(\mathbf{X}\mathbf{W}\bar{\mathbf{x}})]_{i_T}$, and

$$g(w) = \begin{cases} 1, & \text{if } w \geq 0, \\ \frac{1}{e^{-w}}, & \text{otherwise.} \end{cases}$$

The *attention priority similarity* quantifies how well the weights $\mathbf{W}'$ preserve the attention priority of the weights $\mathbf{W}$. A value of 1 indicates that the priority is fully preserved. Using this metric, we

---

[4]*Note:* $\mathbf{W}^{\text{svm}} = \mathbf{0}$ means the number of SCCs is 1 for $\mathcal{G}^k, \forall k \in [K]$. During the simulation, we proceed to the next instance when $\mathbf{W}^{\text{svm}} = \mathbf{0}$ until reaching a total of 100 instances.

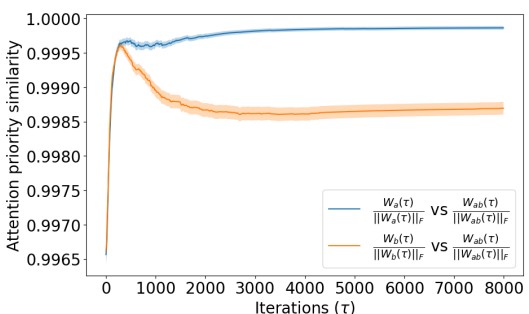

Figure 7: Convergence of attention priority similarity for $\frac{\mathbf{W}_{ab}(\tau)}{\|\mathbf{W}_{ab}(\tau)\|_F}$ relative to $\frac{\mathbf{W}_a(\tau)}{\|\mathbf{W}_a(\tau)\|_F}$ (blue) and $\frac{\mathbf{W}_b(\tau)}{\|\mathbf{W}_b(\tau)\|_F}$ (orange).

Table 1: Proportion of *keep topic*, *ambiguous*, and *change of topic* with varying $T_{test} = \{4, 8, 16, 24, 32, 64, 128, 256, 512\}$.

| $T_{test}$ | KEEP(%) | AMBIGUOUS(%) | CHANGE(%) |
|---|---|---|---|
| 4 | $98.60 \pm 1.54$ | $1.40 \pm 1.54$ | $0.00 \pm 0.00$ |
| 8 | $98.50 \pm 1.47$ | $0.96 \pm 1.11$ | $0.54 \pm 0.76$ |
| 16 | $98.06 \pm 1.33$ | $0.54 \pm 0.76$ | $1.40 \pm 1.07$ |
| 24 | $98.48 \pm 1.31$ | $0.26 \pm 0.44$ | $1.26 \pm 1.10$ |
| 32 | $98.84 \pm 1.15$ | $0.12 \pm 0.33$ | $1.04 \pm 1.03$ |
| 64 | $99.10 \pm 1.07$ | $0.04 \pm 0.20$ | $0.86 \pm 1.05$ |
| 128 | $99.64 \pm 0.53$ | $0.02 \pm 0.14$ | $0.34 \pm 0.52$ |
| 256 | $99.98 \pm 0.14$ | $0.00 \pm 0.00$ | $0.02 \pm 0.14$ |
| 512 | $100.00 \pm 0.00$ | $0.00 \pm 0.00$ | $0.00 \pm 0.00$ |

conduct experiments, with results in Figure 7. We generate embeddings with $K = 10$ and $d = 16$, and randomly construct TPGs for $\mathbb{T}_a$ and $\mathbb{T}_b$. Using these TPGs, we randomly generate DSET$_a$ and DSET$_b$. We compute $\frac{\mathbf{W}_a(\tau)}{\|\mathbf{W}_a(\tau)\|_F}$, $\frac{\mathbf{W}_b(\tau)}{\|\mathbf{W}_b(\tau)\|_F}$ and $\frac{\mathbf{W}_{ab}(\tau)}{\|\mathbf{W}_{ab}(\tau)\|_F}$ using the same procedure as Li et al. [25]. We generate test sequences $\mathbf{Z}$ within $\mathbb{T}_a$, and we calculate the *attention priority similarity* of $\frac{\mathbf{W}_{ab}(\tau)}{\|\mathbf{W}_{ab}(\tau)\|_F}$ relative to both $\frac{\mathbf{W}_a(\tau)}{\|\mathbf{W}_a(\tau)\|_F}$ and $\frac{\mathbf{W}_b(\tau)}{\|\mathbf{W}_b(\tau)\|_F}$. We repeat this process for multiple TPGs and input sequences (simulation details in Appendix C). Figure 7 clearly demonstrates that the similarity converges to 1 after $\tau = 8000$ iterations when evaluated relative to $\frac{\mathbf{W}_a(\tau)}{\|\mathbf{W}_a(\tau)\|_F}$ (blue line), but fails to converge relative to $\frac{\mathbf{W}_b(\tau)}{\|\mathbf{W}_b(\tau)\|_F}$ (orange line). These observations align with the results of Theorem 2.

### C.3 Simulation in Section 5

In Figure 5(a), we predict next tokens for 5000 test sequences from $\mathbb{T}_a$ with $T_{test} = \{4, 8, 16, 24, 32, 64, 128, 256, 512\}$, while fixing $L = 4$, $d = 16$, $T_{train} = 4$, and $K = 10$. The proportion of each scenario with varying $T$ is illustrated in Table 1. For Figure 5(b), we predict next tokens for 5000 test sequences (the sequence length is $T_{test} = 20$) using models trained with $L = \{4, 6, 8, 10, 12, 14, 16, 18\}$, $d = 16$, $K = 10$, and $T_{train} = 4$. The proportion of each scenario with varying $L$ is illustrated in Table 2.

### C.4 Additional experiments for convergence in mixed topics

Building upon the convergence experiments in Li et al. [25], our work demonstrates that the correlation coefficients $\langle \mathbf{W}_{ab}(\tau), \mathbf{W}_{ab}^{svm} \rangle / \langle \|\mathbf{W}_{ab}(\tau)\|_F, \|\mathbf{W}_{ab}^{svm}\|_F \rangle$ (green lines) in Figure 8, measured with varying $K = \{6, 10, 14\}$ and $L = \{8, 12, 16\}$, approach to 1. These results indicate that Theorem 1 extends beyond individual topics to also capture the convergence in mixed-topic scenarios, albeit with relatively slower convergence. In these experiments, we fix $T_{train} = 4$ and $d = 16$. Each point represents the average over 5000 randomly generated instances, trained with 8000 iterations. The shaded area around each line represents the $95\%$ confidence interval, computed over 50 epochs.

Table 2: Proportion of *keep topic*, *ambiguous*, and *change of topic* with varying $L = \{4, 6, 8, 10, 12, 14, 16, 18\}$.

| $L$ | KEEP(%) | AMBIGUOUS(%) | CHANGE(%) |
|---|---|---|---|
| 4 | $98.22 \pm 1.43$ | $0.26 \pm 0.60$ | $1.52 \pm 1.31$ |
| 6 | $98.30 \pm 1.37$ | $0.50 \pm 0.68$ | $1.20 \pm 1.11$ |
| 8 | $98.18 \pm 1.49$ | $0.68 \pm 0.68$ | $1.14 \pm 1.23$ |
| 10 | $98.42 \pm 1.25$ | $0.76 \pm 0.85$ | $0.82 \pm 1.02$ |
| 12 | $98.14 \pm 1.32$ | $0.82 \pm 0.92$ | $1.04 \pm 0.97$ |
| 14 | $97.96 \pm 1.44$ | $1.24 \pm 1.06$ | $0.80 \pm 0.86$ |
| 16 | $98.28 \pm 1.33$ | $0.98 \pm 0.91$ | $0.74 \pm 0.85$ |
| 18 | $98.02 \pm 1.58$ | $1.26 \pm 1.14$ | $0.72 \pm 0.86$ |

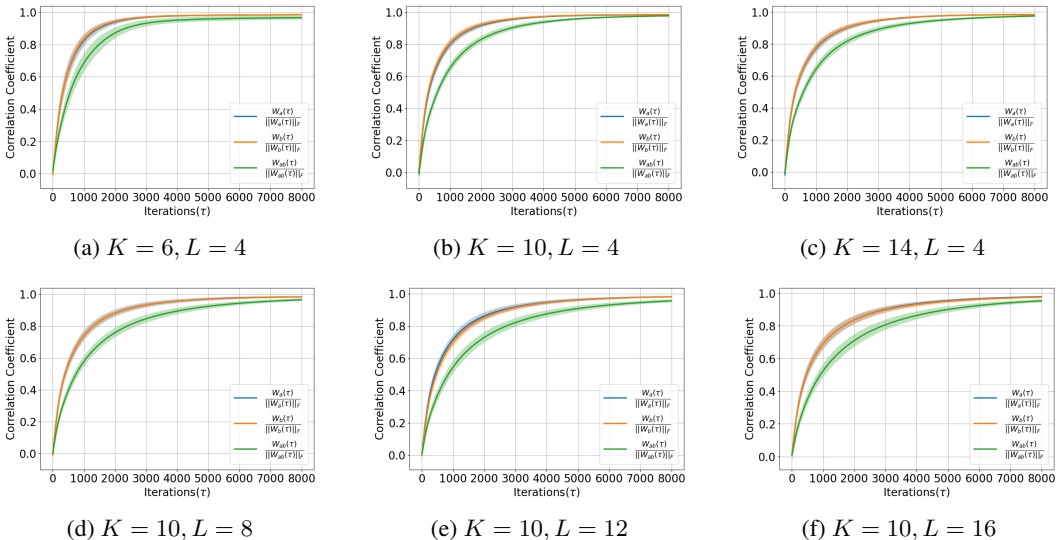

(a) $K = 6, L = 4$     (b) $K = 10, L = 4$     (c) $K = 14, L = 4$

(d) $K = 10, L = 8$     (e) $K = 10, L = 12$     (f) $K = 10, L = 16$

Figure 8: Convergence of $\frac{\mathbf{W}_a(\tau)}{\|\mathbf{W}_a(\tau)\|_F}$ (blue), $\frac{\mathbf{W}_b(\tau)}{\|\mathbf{W}_b(\tau)\|_F}$ (orange), and $\frac{\mathbf{W}_{ab}(\tau)}{\|\mathbf{W}_{ab}(\tau)\|_F}$ (green) for varying $K$ and $L$, with fixed $T_{train} = 4$ and $d = 16$.

### C.5 Numerical analysis for each scenario in Figure 3

Figure 9 provides a numerical breakdown for each scenario in Figure 3. In Figure 9, each distinct color corresponds to a unique token within the input sequence $\mathbf{X}$, which consists of 4 tokens. $\mathbf{e}_4$ is the last token across all three input sequences. For each input sequence $\mathbf{X}$, we apply $\mathbf{W}_a(\tau)$ and $\mathbf{W}_{ab}(\tau)$ with $\tau = 8000$ to predict the next token, yielding $\hat{y}_{\mathbf{W}_a}$ and $\hat{y}_{\mathbf{W}_{ab}}$, respectively.

Let $[\mathbb{S}(\mathbf{X}\mathbf{W}_a(\tau)\bar{\mathbf{x}})]_i = a_i$ and $[\mathbb{S}(\mathbf{X}\mathbf{W}_{ab}(\tau)\bar{\mathbf{x}})]_i = b_i$, for $i \in [T]$. Following Equation 9 and Equation 10, we compute $[Cf_{\mathbf{W}_a(\tau)}(\mathbf{X})]_{x_i}$ and $[Cf_{\mathbf{W}_{ab}(\tau)}(\mathbf{X})]_{x_i}$ to get the *highest probability SCC* and predict the next token for each input sequence.

**Topic continuity.** In Fig. 9a, input sequence $\mathbf{X}$ consists of four unique tokens: $\mathbf{e}_5$, $\mathbf{e}_1$, $\mathbf{e}_3$, and $\mathbf{e}_4$. Based on $\mathcal{G}_a^{(4)}$ in Figure 3 (left), the priority order of these tokens is $\mathbf{e}_5 > \mathbf{e}_3 > \mathbf{e}_1 > \mathbf{e}_4$, with corresponding $a_i$ values: $0.45 > 0.25 > 0.20 > 0.1$. Since $[Cf_{\mathbf{W}_a(\tau)}(\mathbf{X})]_{\mathbf{e}_5} = 1 \times 0.45$ is the largest, $\widehat{\mathcal{G}}_a^{(4)} = \{\mathbf{e}_5\}$ and $\hat{y}_{\mathbf{W}_a} = \mathbf{e}_5$. In the mixed-topic scenario, $\mathbf{W}_{ab}$ preserves the attention priority but $\mathbf{e}_4$ and $\mathbf{e}_1$ have the same priority: $\mathbf{e}_5 > \mathbf{e}_3 > \mathbf{e}_1 = \mathbf{e}_4$, with corresponding $b_i$ values: $0.40 > 0.30 > 0.15 = 0.15$. Token $\mathbf{e}_5$ is still with the highest probability to be chosen, as $[Cf_{\mathbf{W}_a(\tau)}(\mathbf{X})]_{\mathbf{e}_5} = 1 \times 0.40$. Following the Definition 3, $\mathbf{W}_{ab}$ *keeps* topic for the the input sequence $\mathbf{X} = [\mathbf{e}_5, \mathbf{e}_1, \mathbf{e}_3, \mathbf{e}_4]^\top$.

**Ambiguous sequence.** Input sequence $\mathbf{X}$ in Fig. 9b has two unique tokens: $\mathbf{e}_1$ and $\mathbf{e}_4$. The priority order is $\mathbf{e}_1 > \mathbf{e}_4$, following $\mathcal{G}_a^{(4)}$ in Figure 3 (left). The corresponding values are $a_1 = a_3 = 0.3$

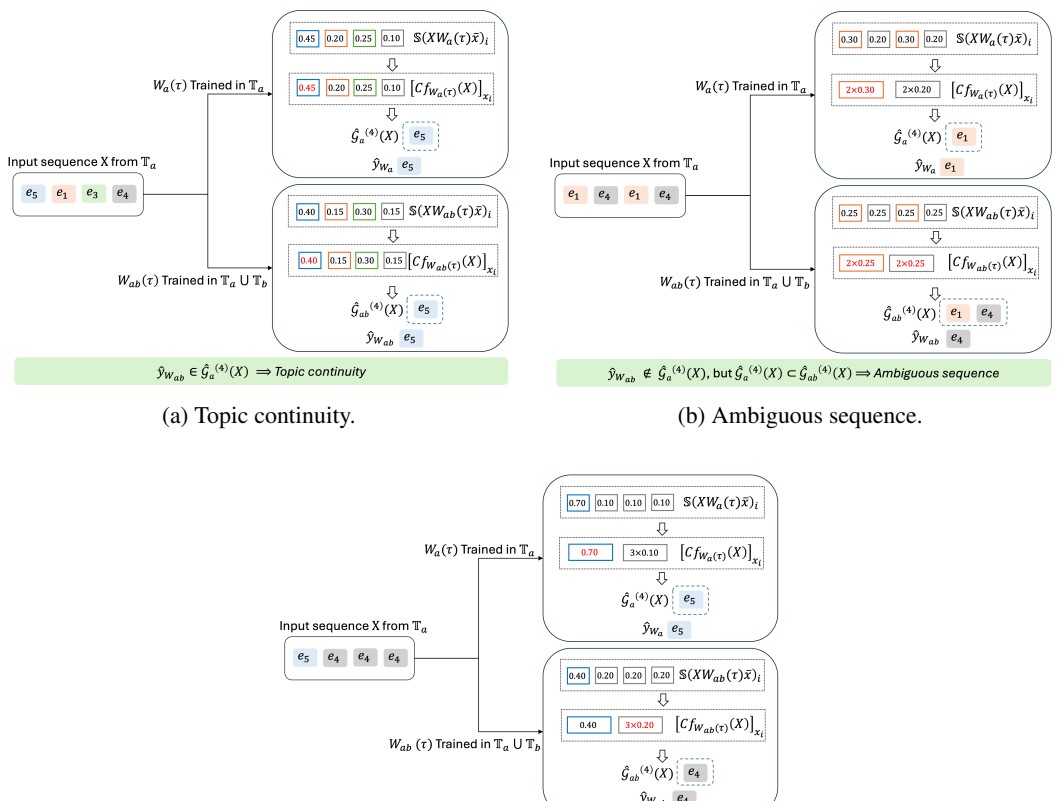

(a) Topic continuity.

(b) Ambiguous sequence.

(c) Change of topic.

Figure 9: Numeric details for each scenario: (a) topic continuity, (b) ambiguous sequence, and (c) change of topic.

and $a_2 = a_4 = 0.2$. Then $[Cf_{\mathbf{W}_a(\tau)}(\mathbf{X})]_{\mathbf{e}_1} = 2 \times 0.30$ and $[Cf_{\mathbf{W}_a(\tau)}(\mathbf{X})]_{\mathbf{e}_4} = 2 \times 0.20$. Thus, $\widehat{\mathcal{G}}_a^{(4)}$ is $\{\mathbf{e}_5\}$ with the highest probability. $\mathbf{W}_{ab}$ makes $\mathbf{e}_4$ and $\mathbf{e}_1$ with the same priority, as indicated by $\mathcal{G}_{ab}^{(4)}$ in Figure 3 (left). Both $\mathbf{e}_1$ and $\mathbf{e}_4$ are within the *highest probability SCC*, $\widehat{\mathcal{G}}_{ab}^{(4)}$, due to $[Cf_{\mathbf{W}_{ab}(\tau)}(\mathbf{X})]_{\mathbf{e}_1} = [Cf_{\mathbf{W}_{ab}(\tau)}(\mathbf{X})]_{\mathbf{e}_4} = 2 \times 0.25$. Although $\hat{y}_{\mathbf{W}_{ab}} \notin \widehat{\mathcal{G}}_a^{(4)}$, $\widehat{\mathcal{G}}_a^{(4)} \in \widehat{\mathcal{G}}_{ab}^{(4)}$. Therefore, the sequence $\mathbf{X} = [\mathbf{e}_1, \mathbf{e}_4, \mathbf{e}_1, \mathbf{e}_4]^\top$ is *ambiguous*, based on the Definition 4.

**Change of topic.** For the input sequence $\mathbf{X}$ in Fig. 9c, the only two unique tokens, $\mathbf{e}_5$ and $\mathbf{e}_4$, are with the same priority order in both $\mathcal{G}_a^{(4)}$ and $\mathcal{G}_{ab}^{(4)}$ from Figure 3 (left): $\mathbf{e}_5 > \mathbf{e}_1$. With $\mathbf{W}_a(\tau)$ trained in $\mathbb{T}_a$, the token $\mathbf{e}_5$ has $a_1 = 0.70$ and the token $\mathbf{e}_4$ has $a_2 = a_3 = a_4 = 0.10$. Obviously, $1 \times 0.70 = [Cf_{\mathbf{W}_a(\tau)}(\mathbf{X})]_{\mathbf{e}_5} > [Cf_{\mathbf{W}_a(\tau)}(\mathbf{X})]_{\mathbf{e}_4} = 3 \times 0.10$. Thus, $\widehat{\mathcal{G}}_a^{(4)}$ consists of $\mathbf{e}_5$. However, $\widehat{\mathcal{G}}_{ab}^{(4)}$ consists of $\mathbf{e}_4$ instead of $\mathbf{e}_5$, due to $1 \times 0.40 = [Cf_{\mathbf{W}_{ab}(\tau)}(\mathbf{X})]_{\mathbf{e}_5} < [Cf_{\mathbf{W}_{ab}(\tau)}(\mathbf{X})]_{\mathbf{e}_4} = 3 \times 0.20$. Since $\hat{y}_{\mathbf{W}_{ab}} \notin \widehat{\mathcal{G}}_a^{(4)}$ and $\widehat{\mathcal{G}}_a^{(4)} \not\subset \widehat{\mathcal{G}}_{ab}^{(4)}$, $\mathbf{W}_{ab}$ *changes topic* for the input sequence $\mathbf{X}$. Moreover, we have $(\mathbf{e}_5 \Rightarrow \mathbf{e}_i) \in \mathcal{G}_{ab}^{(4)}$ for $i \in [4]$, as shown in Figure 3 (left). Thus, the *highest priority SCC* (Definition 1) in $\mathbb{T}_{ab}$ is $\dot{\mathcal{G}}_{ab}^{(4)}(\mathbf{X}) = \{\mathbf{e}_5\}$. In the input sequence $\mathbf{X} = [\mathbf{e}_5, \mathbf{e}_4, \mathbf{e}_4, \mathbf{e}_4]^\top$, the lower-priority token $\mathbf{e}_4 \notin \dot{\mathcal{G}}_{ab}^{(4)}(\mathbf{X})$ appears more frequently than the higher-priority token $\mathbf{e}_5 \in \dot{\mathcal{G}}_{ab}^{(4)}(\mathbf{X})$, illustrating our Theorem 3.

# D Experimental details in Section 6

In this section, we provide the experimental details in four LLMs: GPT-4o, Llama-3.3, Claude-3.7, and DeepSeek-V3. Here, we outline the general procedure used in each model, under identical parameter settings, to generate continuations for each segment of the abstract.

1. Extract the first $T$ words from paper A's abstract as the input segment $\mathbf{X}$ from Topic A.

2. Randomly select 5 papers different from paper A, as papers B in $\{B_i\}_{i=1}^5$.

3. For the input segment $\mathbf{X}$, apply RAG to extract top 3 relevant excerpts (chunks) from paper A as the the knowledge A, denoted as $\text{Ref}_A$. Each chunk has 800 tokens length.

4. Similarly, retrieve top 3 relevant excerpts from paper $B_i$ as the knowledge $B_i$, denoted as $\text{Ref}_{B_i}$.

5. Combine the knowledge from Topic A and from Topic $B_i$ as the knowledge $AB_i$ for mixed Topics, denoted as $\text{Ref}_{AB_i}$.

6. For the input segment $\mathbf{X}$, promot each LLM with $\text{Prompt}_A$ and $\text{Prompt}_{AB_i}$, to generate the continuations as $\hat{y}_{\mathbf{W_a}}$ and $\hat{y}_{\mathbf{W_{ab}}}$, respectively. Notably, the only difference between $\text{Prompt}_A$ and $\text{Prompt}_{AB_i}$ is the reference excerpts provided $\text{Ref}_A$ or $\text{Ref}_{AB_i}$. All LLMs are set with a temperature of 0 to match the greedy decoding in our theoretical framework. The maximum completion length was set to 1000 tokens to ensure that the generated continuations could complete the abstract.

    (a) $\text{Prompt}_A$:
    *Here are some relevant excerpts from research paper(s) as reference:$Ref_A$. Below is the 1st fragment of an abstract from arXiv paper A: $\mathbf{X}$. Please continue the 2nd fragment of the abstract based on the relevant excerpts without including the given content in the output.*

    (b) $\text{Prompt}_{AB_i}$:
    *Here are some relevant excerpts from research paper(s) as reference:$Ref_{AB_i}$. Below is the 1st fragment of an abstract from arXiv paper A: $\mathbf{X}$. Please continue the 2nd fragment of the abstract based on the relevant excerpts without including the given content in the output.*

7. Calculate the average cosine similarity between $\hat{y}_{\mathbf{Wa}}$ and $\hat{y}_{\mathbf{Wab}}$ across five pairs of paper A and paper $B_i$.

## D.1 Impact of input length

To investigate the impact of the input length, we vary $T = \{10, 30, 50, 70, 90, 110, 130, 150\}$ for every paper as Topic $A$, increasing the length of the input segment $\mathbf{X}$ extracted from the abstract of paper A, as shown on the x-axis from Figure 5a.

## D.2 Impact of topic ambiguity

We quantify topic (paper) ambiguity by computing the average similarity among each paper's keywords. Since arXiv papers do not provide keywords, we use Llama-3.3 to generate four keywords for each paper prior to generating continuations with the LLMs. To investigate the topic ambiguity, we fix the input length with $T = 80$ for every paper as Topic A and order papers by the average keywords similarity, as shown on the x-axis of Figure 5b. A higher keywords similarity corresponds to lower topic ambiguity.

## D.3 Additional experiments

### D.3.1 Extended input length

To further examine the impact of the input length, we randomly select 50 out of 100 arXiv papers introduced in Sec. 6 and extend the input length with the first $310, 460, \ldots, 1210$ words from the introduction of each paper A as the input prompt. We start from $T = 310$ because shorter introductions provide insufficient contextual information to reliably extract relevant excerpts from the

full paper when using RAG. As shown in Figure 10, the average cosine similarity generally increases with the more introduction content from paper A, with the exception of DeepSeek, which exhibits only a marginal improvement.

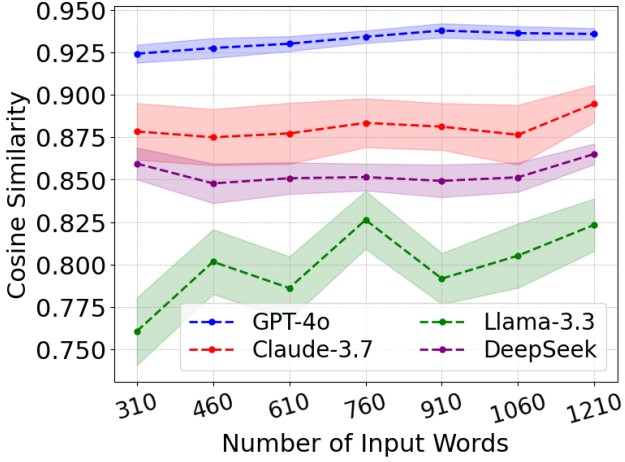

Figure 10: Extended length of input sequences.

### D.3.2 Alternative measure of ambiguity

As an alternative approach to validate our results, we also measure the ambiguity based on the similarity between the keywords of paper A and those of paper B. Following the same setup, we rank the cosine similarity for each paper A by the increasing similarity between its keywords and those of five papers B, where a higher level corresponds to a greater ambiguity. As shown in Figure 11, our results still hold since the cosine similarity remains relatively stable as the ambiguity level increases.

## E   Computational resources for experiments

In our simulations based on the single-layer self-attention model, each group of parameter setting requires 7 hours to train two models separately, one for single input topic and one for mixed topics, followed by 2 additional hours for next-token prediction.

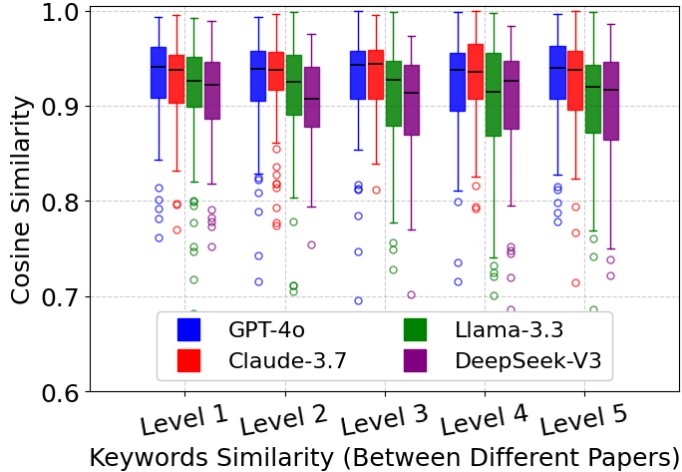

Figure 11: Topic ambiguity measured by the similarity between the keywords of paper A and those of paper B. Higher keywords similarity indicate greater topic ambiguity between the two papers.

In our experiments on LLMs, we query GPT-4o, Llama-3.3, Claude-3.7, and DeepSeek-V3 through API calls. All experiments were conducted on a standard laptop without specialized hardware. For each LLM, the full process, including selecting relevant excerpts using RAG and generating continuations, requires approximately 80 hours of runtime, with a total of 30 million input tokens and 5 million output tokens. The total API usage cost for the experiments is approximately 350 USD.

## F  Impact statement

Our investigation highlights fundamental differences between spontaneous topic changes in LLMs and spontaneous human thought, informing the development of more natural and flexible AI systems in domains such as customer service and mental health support. However, improving such capabilities can raise ethical considerations, including inadvertent manipulation of user focus, especially in persuasive or sensitive contexts. Our work, while largely theoretical, emphasizes the importance of fairness, privacy, and user autonomy as developers refine these systems to serve users' interests, respect contextual boundaries, and remain accountable. This research has the potential to advance both Machine Learning and Human-Computer Interaction by informing new architectures that mimic human-like topic shifts; nevertheless, any real-world application of these findings should be accompanied by vigilant oversight to mitigate risks of misuse—such as deceptive or manipulative dialogue shifting. There are many other potential societal consequences of our work, none which we feel must be specifically highlighted here.

