# OpenReview forum: "Dynamics of Spontaneous Topic Changes in Next Token Prediction with Self-Attention"
_NeurIPS.cc/2025/Conference — NeurIPS 2025 poster_

### Official Review · Reviewer_2tA4 · 2025-07-01

**Clarity:** 1
**Significance:** 2
**Originality:** 3
**Rating:** 4
**Confidence:** 3

**Summary:**

This paper aims to propose a theoretical model of ‘spontaneous topic changes in autoregressive self-attention models, motivated by observation that spontaneous topic change as a fundamental cognitive feature in humans while LLMs topic change is rather initiated by contextual cue. Using a concept of topic priority graphs (TPGs), they propose a model of such topic change in a simplified 1-layer self-attention language model and provide empirical results on the frontier LLMs that capture the prediction of their model about input length and topic ambiguity.

**Questions:**

Line 310: isn’t it the opposite; lower similarity of the topics implies less overlap between A and B?

I thing Figure 5 a is supporting the authors claim rather weakly; have you increased the context length further to see the effect better? Does your theory provide any tendency of the topic change probability change (linear, or sudden change)?

Have you considered other metric of topic similarity, which might be able to capture more intricate relationship? (eg. hierarchy)

**Ethical Concerns:**

["NO or VERY MINOR ethics concerns only"]

**Final Justification:**

Authors addressed the questions well, which led to an increase in score from 3 to 4. I appreciated the additional experiments on longer context, and I trust that the authors will update the corresponding bits of the paper for clarity + discussion of future work.

**Limitations:**

The theory is limited to a simple setup but the authors already mention it.

While authors motivate the work with the dissimilarity between human cognition and LLMs in topic change, I do not see enough evidence of it.

The definition of similarity seems to be limited - what about hierarchy within topics or high dimensional topics?

**Quality:**

3

**Strengths And Weaknesses:**

## Strengths:

I think it is valuable to make a quantitative approach based on a simplified model to capture and describe phenomena in many complex models. Also, the authors provide the experiments with the frontier models at scale which captures some aspect of the simplified model, which validates their theory.

## Weaknesses:

The biggest weakness of the paper is clarity. I found the use of notations hard to follow and it was hard to follow the path to the ultimate end prediction about input length and topic ambiguity.
The visual illustration of the model is unclear; for example, it is not clearly described what is the direction of the edges and what is the dotted square. I inferred that it’s a label to input and SCCs but it took a lot of effort for me. Having explicit description of the figure would improve readability.

While authors motivate the study by claiming the difference between human cognition and LLM on spontaneous topic change, I think its basis is not very well supported.

---

> ### Author Rebuttal · Authors · 2025-07-28
>
> Thank you for acknowledging the value of our quantitative approach and the inclusion of real LLMs experiments.
> Let us first explain the basis of our motivations:
>
> Our research seeks to understand both the similarities and the differences between AI and human cognition. In the Introduction (line 22), we reference prior work on the area of spontaneous thought, which implies that spontaneous thought is natural and frequent for humans. Neuroscientists model spontaneous thoughts as traversals across conceptual graphs, where the mind jumps between related concepts or topics. Our framework mirrors this process, aiming to explore what drives spontaneous topic changes in LLMs. Our aim is to draw a parallel between spontaneous topic changes in LLMs and spontaneous human thought, examining the phenomenon from both theoretical and empirical perspectives and giving preliminary steps towards answering a foundational question "what truly distinguishes human thought from AI?. To the best of our knowledge, our work is among the first to address this question through a combination of theoretical analysis and experimental validation.
>
> Now let us clarify some of your questions that will help supporting our conclusions:
>
> - *Line 310: isn’t it the opposite; lower similarity of the topics implies less overlap between A and B?*
>
> We believe there may be a misunderstanding. In Section 6 (line 309), we quantify topic ambiguity by keyword similarity within a single paper, not keywords from two or more papers. For instance, a higher keywords similarity within Paper A indicates a more coherent topic in Paper A, leading to less probability of overlap between paper A and others and thus implying the lower topic ambiguity for Paper A. Put simply, if a paper’s keywords are very diverse, it suggests that the paper spans multiple topics, increasing the likelihood of overlap with other papers and thus leading to greater ambiguity. We will clarify this explanation in the revision to avoid potential confusion.
>
> - *I thing Figure 5 a is supporting the authors claim rather weakly; have you increased the context length further to see the effect better?*
>
> We appreciate this comment. We intentionally kept the context length small for simplicity, but our experimental setup can be readily extended to longer contexts, further supporting our claims. In fact, below we provide a summary of real experimental results using GPT-4o with input lengths ranging from 0 to 1000 tokens. While we are unable to include figures, we hope the upward trend remains evident from the data provided.
>
> | Input words | Cosine similarity |
> |---|---|
> | 10 |  0.877 |
> | 210 | 0.915 |
> | 410 | 0.919 |
> | 610 | 0.927 |
> | 810 | 0.926 |
> | 1010 | 0.935 |
>
> We can easily extend Figure 5a to 1000+ words in the final manuscript with all LLMs.
>
> - *Does your theory provide any tendency of the topic change probability change (linear, or sudden change)?*
>
> Our theoretical results show that the probability of topic changes vanishes as the input sequence length increases sufficiently and does not increase as the topic ambiguity increases. Our goal in this theorem is to show the difference between LLMs and human in spontaneous topic changes. We didn’t provide the specific tendency of the probability, but we thank you for your thoughtful question.
>
> - *Have you considered other metric of topic similarity, which might be able to capture more intricate relationship? (eg. hierarchy)*
>
> As an initial step, our current framework aims to address when and why self-attention models preserve the topic or spontaneously change to another. While our work doesn’t model topic hierarchy, your insightful question offers a promising direction for future work, bridging spontaneous topic changes with hierarchical semantic structures.
>
> - Comments on visual illustration not being clear
>
> As correctly inferred, the direction of edge is from output to input (see line 114), and the dotted square is SCC. We acknowledge our unclear visual illustration and thank you for pointing this out. We’ll clarify these details in the caption to improve readability.
> ___________________
> To conclude, we believe the reason for not receiving a higher rating may reflect some hesitation to see how our experiments and conclusions support our core motivation. We hope that our clarification on the notion of ambiguity, along with the additional validation using longer input lengths, helps reinforce the validity of our main conclusion: *unlike human cognition, increasing context length or input ambiguity does not make spontaneous topic change more likely in LLMs*. This insight opens a path toward addressing the broader question: How does human cognition differ from AI? We see this direction as closely aligned with NeurIPS’ mission to promote meaningful, thought-provoking exploration in machine learning.

---

> > ### Author Response · Authors · 2025-08-05
> >
> > Following up on your question regarding topic ambiguity, as well as a related comment from Reviewer L1rU, we have generated an additional result that we believe further clarifies our claims. Please refer to our latest author comment under Reviewer L1rU for details.

---

> > > ### Comment · Reviewer_2tA4 · 2025-08-07
> > >
> > > I appreciate the authors responses and have updated my score correspondingly. I think some discussion of future directions (e.g., related to hierarchy) could benefit the paper/discussion and hope the authors adopt their paper accordingly.

---

> > > > ### Author Response · Authors · 2025-08-07
> > > >
> > > > Thank you very much for your valuable inputs and for updating your score. We appreciate your suggestions and will include an expanded discussion on future directions in the final manuscript.

---

### Official Review · Reviewer_V5wh · 2025-07-02

**Clarity:** 3
**Significance:** 2
**Originality:** 3
**Rating:** 4
**Confidence:** 3

**Summary:**

This paper investigates spontaneous topic changes, i.e., abrupt, unprompted shifts in conversational topic, within self-attention-based language models and reveals their differences from spontaneous thought patterns seen in human cognition. The authors define a formal framework using token-priority graphs (TPGs) to model topic representation and transition dynamics in simplified self-attention settings. Their results indicate that self-attention-based language models rarely change topics spontaneously unless specific statistical conditions are met, such as when low-priority tokens dominate. Contrary to human behavior, longer inputs or higher topic ambiguity reduce the likelihood of topic change. Experiments on modern LLMs (e.g., GPT-4o, Llama, Claude) empirically support these theoretical findings.

**Questions:**

Please see the weaknesses in the "Strengths And Weaknesses" section.

**Ethical Concerns:**

["NO or VERY MINOR ethics concerns only"]

**Final Justification:**

Considering the limited practical utility of the proposed method as currently presented, I will maintain my current score.

**Limitations:**

yes

**Paper Formatting Concerns:**

There is no formatting concerns.

**Quality:**

3

**Strengths And Weaknesses:**

**Strengths**:
1. The paper explores an interesting and novel research question: the spontaneous topic change within LLMs and how it differs from human spontaneous thought.
2. A rigorous mathematical framework is developed using TPGs and SVM-based analysis to explore the conditions under which spontaneous topic changes can or cannot occur in self-attention models.
3.  Experimental results on real-world LLMs (GPT-4o, Claude-3.7, LLaMA-3.3, and DeepSeek-V3 ) empirically confirm the theoretical predictions.

**Weaknesses**:
1. Despite the paper tackles an interesting phenomenon: the spontaneous topic change within LLMs, it lacks a clear justification for its practical significance, leaving readers uncertain about why understanding or modeling such behavior in LLMs is important for real-world applications or cognitive alignment.

2. The condition under which a topic change occurs: when a lower-priority token appears more frequently than all higher-priority tokens, is theoretically well-defined but practically hard to verify, since the paper’s formalism relies on constructing token-priority graphs that are not directly accessible or interpretable in real-world settings. Moreover, the paper does not provide evidence or discussion on how frequently such conditions arise in realistic language model usage.

3. The paper focuses on analyzing the conditions for spontaneous topic change but does not explore potential applications of its findings, such as methods to encourage or suppress such behavior in language models for specific tasks.

---

> ### Author Rebuttal · Authors · 2025-07-28
>
> Thank you for comments. See here our reply:
>
> - *Despite the paper tackles an interesting phenomenon: the spontaneous topic change within LLMs, it lacks a clear justification for its practical significance, leaving readers uncertain about why understanding or modeling such behavior in LLMs is important for real-world applications or cognitive alignment.*
>
> Our research seeks to understand both the similarities and the differences between AI and human cognition.
> Although its immediate value is not an engineering deliverable, the work gives preliminary steps towards answering a foundational question: what truly distinguishes human thought from AI? Cognitive science characterizes spontaneous thought as a distinctly human trait. Neuroscientists model spontaneous thoughts as traversals across conceptual graphs, where the mind jumps between related concepts or topics. Our framework mirrors this process, aiming to explore what drives spontaneous topic changes in LLMs. Our aim is to draw a parallel between spontaneous topic changes in LLMs and spontaneous human thought, examining the phenomenon from both theoretical and empirical perspectives.
>
> Specifically, understanding that longer inputs reduce the probability of topic changes has practical significance. Especially for the long-form question answering (multi-paragraph in response) for scientific or technical tasks, providing longer input contexts helps reduce unintended topic changes, leading to more coherent outputs.
>
> To the best of our knowledge, our work is among the first to address this question through a combination of theoretical analysis and experimental validation.
>
> - *The condition under which a topic change occurs: when a lower-priority token appears more frequently than all higher-priority tokens, is theoretically well-defined but practically hard to verify, since the paper’s formalism relies on constructing token-priority graphs that are not directly accessible or interpretable in real-world settings.*
>
> Thank you for acknowledging our well-defined claim. Our framework is based on Token Priority Graphs (TPGs), where a topic is defined as a set of TPGs. While this doesn't map one-to-one with real-world topics, it offers valuable insights. For instance, one could interpret a topic as a graph of ideas, where certain ideas consistently hold higher priority (a perspective aligned with how some neuroscientists model human thought). Under this view, our condition that “a lower-priority token appears more frequently than all higher-priority tokens” can be seen as a lower-priority idea dominating discourse. This remains a hypothesis, but it opens several promising avenues for future research.
>
> - *The paper focuses on analyzing the conditions for spontaneous topic change but does not explore potential applications of its findings, such as methods to encourage or suppress such behavior in language models for specific tasks.*
>
> Following up with our previous comments. Our work focuses on the simplified single-layer self-attention architecture. This controlled setting allows us to formalize the dynamics of spontaneous topic changes and draw parallels between human cognition and LLMs. While we don’t explore potential applications and frequency in real-world LLMs, our findings lay an important groundwork for future directions.
> ___________
>
> To conclude, we believe the reason for not receiving a higher rating may reflect some hesitation to view our contribution beyond its practical engineering implications. Our work represents foundational research aimed at addressing non-traditional questions that extend beyond typical engineering concerns. We see this direction as highly aligned with NeurIPS’ mission of fostering meaningful, thought-provoking exploration in machine learning.

---

### Official Review · Reviewer_L1rU · 2025-07-02

**Clarity:** 3
**Significance:** 3
**Originality:** 3
**Rating:** 4
**Confidence:** 4

**Summary:**

This paper presents a theoretical and empirical study of spontaneous topic changes in self-attention models, contrasting their behavior with the dynamics of spontaneous human thought. The authors develop a formal framework by defining a "topic" as a set of Token-Priority Graphs (TPGs) within a simplified single-layer self-attention model. This approach leads to several key findings that distinguish the mechanisms of large language models (LLMs) from human cognition: 1) Preservation of topic priorities: The model preserves the priority order of tokens related to an input's original topic, even when trained on a mix of different topics. 2) Conditions for topic change: A topic shift is shown to occur only under a specific condition: a lower-priority token must appear more frequently in the input than all higher-priority tokens associated with the current topic. 3) Impact of context and topic ambiguity: Contrary to human thought, where longer discussions or ambiguity can trigger new associations, the paper establishes that longer input contexts decrease the likelihood of a topic change in LLMs. Similarly, greater ambiguity between topics acts as a stabilizing factor rather than a trigger for spontaneous shifts

**Questions:**

- Clarity of definitions: Some symbols are used without definition. For instance, $\mathbb{Z}$ in Assumption 3.

- Justification for the topic ambiguity metric: In Section 6, the authors mention that a higher keywords similarity between two papers corresponds to lower topic ambiguity. This is counterintuitive as a higher overlap in the keywords of two papers generally implies higher topic ambiguity.

**Ethical Concerns:**

["NO or VERY MINOR ethics concerns only"]

**Final Justification:**

Most of my concerns have been resolved so I'm happy to raise my score from 2 to 4 and recommend for acceptance.

**Limitations:**

Limitations and potential societal impacts are discussed in detail.

**Quality:**

3

**Strengths And Weaknesses:**

Strengths:
- The paper is well-motivated by the goal of theoretically understanding the distinctions between LLMs and human cognition.

Weaknesses:
- Limited discussion of literature on topic shifts: While the authors mention several papers on spontaneous topic changes in LLMs in the introduction part, a more detailed discussion comparing the proposed theoretical framework against existing work is needed.

- Flawed application of prior theoretical results. The paper's theoretical claims are heavily dependent on Theorem 1, which is adapted from Theorem 2 in Li et al. (2024). However, there appears to be a critical misunderstanding in applying the theorem, which invalidates the subsequent proofs.
  - Informally speaking, the work by Li et al. establishes that the attention weight $W$ converges to a structure that can be decomposed into two components, i.e., $W \approx C \cdot W^{svm} + W^{fin}$ with $C \to \infty$. The two components have different functionality: 1) Hard retrieval ($W^{svm}$): This term selects the high-priority input tokens according to the token-priority graph. 2) Soft composition ($W^{fin}$): A finite component that governs how to assign probabilities among the high-priority tokens selected by $W^{svm}$.

   - However, in this paper, the authors only discuss the impact of $W^{svm}$, ignoring the role of $W^{fin}$. This omission is critical as it leads to incorrect conclusions. E.g., Lemma 1 builds an equivalence, stating that tokens have the same prediction probability if and only if they are in the same strongly connected component (SCC) in the token-priority graph. This is not true without knowing $W^{fin}$ which can (and generally will) assign different probabilities to tokens even if they are in the same SCC.


[1] Yingcong Li, Yixiao Huang, Muhammed E Ildiz, Ankit Singh Rawat, and Samet Oymak. Mechanics of next token prediction with self-attention. In International Conference on Artificial Intelligence and Statistics, pages 685–693. PMLR, 2024.

---

> ### Author Rebuttal · Authors · 2025-07-28
>
> Thank you for your comments.
>
> **Weaknesses**
> 1. We would like to begin by clarifying that we are confident in the correctness of our theoretical results under the assumptions we state, (and those from Theorem 2 in Li et al). We believe the confusion may stem from the perception that our analysis should involve the behavior of  $W_{fin}$, but let us explain why this is not needed for our analysis.
>
> To clarify:
>
> The core idea behind Theorem 2 of Li et al. (and its informal version in their Theorem 1) is that self-attention can be decomposed into a *hard retrieval* component ($W_{hard}$ or $W_{svm}$) which selects the highest-priority tokens, and a *soft composition* component ($W_{soft}$ or $W_{fin}$) which determines the probabilities among the selected tokens. Our analysis focuses specifically on the hard retrieval mechanism  ($W_{hard}$ or $W_{svm}$), that is: which tokens are selected as high-priority. Our three key definitions (*keep topic*, *ambiguous sequence*, *change of topic*) are based entirely on this component, therefore the analysis on $W_{fin}$ is not needed. We emphasize $W_{svm}$ in particular to preserve the conceptual parallel with the graph-based prioritization of ideas observed in models of human cognition.
>
> In particular:
>
> - For our Theorem 2, our results concern the preservation of the highest-priority tokens in a combined-topic setting. Since this theorem is on weights $\tilde{W}$ which comes from $W_{svm}$ (*hard retrieval*, no need of $W_{fin}$), the behavior of $W_{fin}$ is irrelevant to the proof. Our conclusions remain valid.
>
> - For Theorems 3 and 4, the definitions (*keep topic*, *ambiguous sequence*, *change of topic*)  depend solely on the highest-probability SCC which is based on the *hard retrieval* component. Again, no dependence on $W_{fin}$  is required, and the results are unaffected.
>
> - Lemma 1 applies to $\tilde{W}$ which comes from $W_{svm}$ (*hard retrieval*, no need of $W_{fin}$). We agree the conclusion of the lemma could be misleading and we will change line 144 to "This means that the tokens that maximize the probability for weights $\tilde{W}$ in Equation 1 are all within the same SCC".
>
> - Our formal proofs are provided in detail in the appendix, and the accompanying numerical simulations illustrate and support their validity.
>
> We appreciate this comment and will revise the paragraph on line 133 to clarify that our focus is on the *hard retrieval* component of Li et al., as the *soft composition* step is not required for our definitions or theoretical results.
> We believe this may be the primary reason behind the reviewer's lowered rating and we trust this explanation resolves the misunderstanding and confirms the validity of our contributions.
>
> 2. Regarding the comment on the limited discussion, we thank the reviewer for the suggestion. While numerous theoretical frameworks exist for analyzing topic shifts in text, to the best of our knowledge, this is the first work to formally connect spontaneous topic changes in large language models (LLMs) with analogous phenomena in human thought, which is why we focused the discussion on this aspect.
>
> **Questions**
>
> - *Justification for the topic ambiguity metric: In Section 6, the authors mention that a higher keywords similarity between two papers corresponds to lower topic ambiguity. This is counterintuitive as a higher overlap in the keywords of two papers generally implies higher topic ambiguity.*
>
> We believe there may be a misunderstanding. In Section 6 (line 309), we quantify topic ambiguity by keyword similarity within a **single** paper, not keywords from two or more papers. For instance, a higher keywords similarity within Paper A indicates a more coherent topic in Paper A, leading to less probability of overlap between paper A and others and thus implying the lower topic ambiguity for Paper A. Put simply, if a paper’s keywords are very diverse, it suggests that the paper spans multiple topics, increasing the likelihood of overlap with other papers and thus leading to greater ambiguity. We will clarify this explanation in the revision to avoid potential confusion.
>
> - *Clarity of definitions: Some symbols are used without definition. For instance, $\mathbb{Z}$ in Assumption 3.*
>
> Thank you for pointing this out. We tried to use standard notation, here $\mathbb{Z}$ is the standard symbol for a set of integers.
> _______________
>
> To conclude, we believe we are pioneering an underexplored area by drawing parallels between spontaneous human thought and spontaneous topic changes in self-attention models. To the best of our knowledge, this is the first study to examine these questions by investigating key similarities and differences between human cognition and AI models in the area of spontaneous thought. We view this foundational research as highly relevant to NeurIPS' mission of advancing meaningful, thought-provoking exploration in machine learning.

---

> > ### Comment · Reviewer_L1rU · 2025-08-01
> > **Response to the rebuttal**
> >
> > Thank you for your responses—most of my concerns have been addressed. I have just a couple of additional comments:
> >
> > - Notation of $\tilde{W}$ vs. $W$
> >
> > I was initially confused by the mixed use of $\tilde{W}$ and $W$ (for example, in line 142 it looks like $\tilde{W}$ was intended). I recommend inserting a brief remark immediately after Theorem 1 to clarify that all subsequent analysis refers to the directional component $\tilde{W}$.
> >
> > - Token-ambiguity metric
> >
> > While my confusion about the metric itself is now resolved, could you explain why you chose this particular formulation rather than computing keyword similarity directly between paper A and paper B? I found Reviewer 2tA4 also has a similar confusion. From first principles, the latter seems more intuitive—would your experimental conclusions still hold under that alternative?

---

> > > ### Author Response · Authors · 2025-08-02
> > >
> > > We are glad that our previous responses addressed most of your concerns.
> > >
> > > - Notation of $\tilde{W}$ vs. $W$
> > >
> > > You are right, there is a typo in line 142 (it should be $\tilde{W}$), sorry for this. In the proof in Appendix A.1 it is shown as it should be. We will revise and insert the remark as you suggest after Theorem 1.
> > >
> > > - Token-ambiguity metric
> > >
> > > For the token-ambiguity metric, our original approach to quantify the ambiguity within each Paper A is to closely follow the theoretical framework in Theorem 4.2. The Theorem claims that as the number of edges $L$ in Topic’s TPGs increases, the probability of topic changes doesn’t increase. Clearly, the more edges, the higher the ambiguity. By analogy, the higher keywords similarity within Paper A indicates greater ambiguity in topic A, and therefore our simulation.
> > >
> > > But based on your suggestion, we generated a new report measuring ambiguity as the similarity between keywords in paper A and B (see table below for gpt-4o since we aren't allowed to show pictures). In the table, the higher level, the higher ambiguity. As you can observe, our results still hold since the cosine similarity remains relatively stable, even as the ambiguity level increases.
> > >
> > > | Ambiguity level | Cosine similarity |
> > > |------|------|
> > > | Level 1 | 0.929 |
> > > | Level 2 | 0.921 |
> > > | Level 3 | 0.928 |
> > > | Level 4 | 0.925 |
> > > | Level 5 | 0.929 |
> > >
> > > We can easily include these results in the manuscript for all LLMs.
> > >
> > > We trust these explanations resolve the misunderstanding and confirm the validity of our contributions.

---

> > > > ### Comment · Reviewer_L1rU · 2025-08-04
> > > >
> > > > Thank you for the response. My concerns have been fully addressed, and I am happy to raise my score.

---

> > > > > ### Author Response · Authors · 2025-08-07
> > > > >
> > > > > We are glad that the reviewer’s concerns have been fully addressed. Thank you very much for your valuable inputs and for increasing the score.

---

### Author Response · Authors · 2025-08-07
**General Response**

We highly appreciate the reviewers’ insights and efforts. We believe we are pioneering an underexplored area by drawing connections between **spontaneous human thought** and **spontaneous topic changes in self-attention models**.

We are encouraged by the reviewers’ recognition of several strengths in our work:

- The paper is well-motivated (*L1rU, V5wh*);
- A rigorous and valuable mathematical framework is developed (*V5wh, 2tA4*);
- Experiments result on frontier models empirically validate the theory (*V5wh, 2tA4*).

We now summarize main concerns raised by reviewers and how we addressed them:

- Significance

To clarify the **novelty and significance** of our contribution: This is, to our knowledge, the first work that (1) **theoretically analyzes** when and why self-attention models allow spontaneous topic changes, and (2) **empirically demonstrates** that these results remain in modern large language models (LLMs), underscoring a clear divergence from spontaneous human cognition. That combination of **formal theory**, applied to the topic dynamics of self-attention, with **empirical validation** in modern models, distinguishes our work and opens the path into the intersection of AI and human cognition.

In an era of growing concern about the similarities between AI and human cognition, our framework proposes a rigorous, interpretable distinction between the two, opening pathways for future research at the intersection of AI and cognition

Specifically, longer inputs reduce the probability of topic changes, providing an insight with practical significance for long-form question answering.

- Clarity and validity of our results

We believe we have thoroughly addressed Reviewer *L1ru*’s concerns regarding the theoretical soundness of our work. In addition, we have made a concerted effort to respond to the general and experimental feedback provided by Reviewers *2tA4* and *V5wh*.
We have extended our results on frontier models to address longer contexts and topic ambiguity, as discussed in our responses to Reviewers *2tA4* and *L1ru*, respectively. These findings can be readily incorporated into the final manuscript, including evaluations across all reference LLMs, thereby strengthening the applicability and relevance of our theoretical contributions.
We will also clarify the caption of Figure 3 to improve readability and correct minor typographical issues, particularly those that may have caused confusion between $\tilde{W}$ and $W$.

Overall, we believe our contribution is novel, original, and holds strong potential for cross-disciplinary impact, well aligned with NeurIPS’s mission to publish transformative and forward-looking research.

---

### Decision · Program_Chairs · 2025-09-17

**Decision:**

Accept (poster)

**Comment:**

This paper examines spontaneous topic changes in self-attention models and compares them to the dynamics of spontaneous human thought. The reviewers agree that the topic is interesting and that the results are solid. However, they also note that the theoretical analysis is conducted in a highly simplified setting, which may limit the practical impact of the findings. After discussion, the authors have satisfactorily addressed several previous concerns raised by the reviewers. All reviewers now support accepting the paper. I concur with the reviewers’ assessments and recommend acceptance.